



# The RHOSSA campaign: Multi-resolution monitoring of the seasonal evolution of the structure and mechanical stability of an alpine snowpack

Neige Calonne[1,2*], Bettina Richter[1*], Henning Löwe[1], Cecilia Cetti[1], Judith ter Schure[1], Alec Van Herwijnen[1], Charles Fierz[1], Matthias Jaggi[1], and Martin Schneebeli[1]

[1]WSL Institute for Snow and Avalanche Research SLF, Davos Dorf, Switzerland
[2]now at Météo-France – CNRS, CNRM UMR 3589, Centre d'Etudes de la Neige, Grenoble, France
[*]These authors contributed equally to this work

**Correspondence:** Neige Calonne (neige.calonne@meteo.fr)

**Abstract.** The necessity of characterizing snow through objective, physically-motivated parameters has led to new model formulations and new measurement techniques. Consequently, essential structural parameters such as density and specific surface area (for basic characterization) or mechanical parameters such as the critical crack length (for avalanche stability characterization) gradually replace the semi-empirical indices acquired from traditional stratigraphy. These advances come along with new demands and potentials for validation. To this end, we conducted the RHOSSA field campaign, in resemblance of density ($\rho$) and specific surface area (SSA), at the Weissfluhjoch research site in the Swiss Alps to provide a multi-instrument, multi-resolution dataset of density, SSA, and critical crack length over the complete winter season 2015-2016. In this paper, we present the design of the campaign and a basic analysis of the measurements alongside with predictions from the model SNOWPACK. To bridge between traditional and new methods, the campaign comprises traditional profiles, density cutter, Ice-Cube, SnowMicroPen (SMP), micro-computed-tomography, propagation saw tests, and compression tests. To bridge between different temporal resolutions, the traditional weekly to bi-weekly snow pits were complemented by daily SMP measurements. From the latter, we derived a re-calibration of the statistical retrieval of density and SSA for SMP version 4 that yields an unprecedented, spatio-temporal picture of the seasonal evolution of density and SSA in a snowpack. Finally, we provide an inter-comparison of measured and modeled estimates of density and SSA for 4 characteristic layers over the entire season to demonstrate the potential of high temporal resolution monitoring for snowpack model validation.

## 1 Introduction

Regular snow monitoring programs are one of the cornerstones of snow science providing valuable time-series of snow properties (e.g. Reba et al., 2011; Morin et al., 2012; Landry et al., 2014; Wayand et al., 2015; Leppänen et al., 2016; Lejeune et al., 2019). Such time-series are indispensable for the development and evaluation of snow models (e.g. Fierz, 1998; Etchevers et al., 2004; Morin et al., 2013; Essery et al., 2016; Krinner et al., 2018) as well as for various applications such as snowpack stability assessment for avalanche risk forecasting (e.g. Schweizer and Wiesinger, 2001; van Herwijnen and Jamieson, 2007),



snowpack processes studies (e.g. Dumont et al., 2017), snow property retrievals from remote sensing (e.g. Leinss et al., 2016; King et al., 2018), water resources estimations (e.g. Jonas et al., 2009), climate studies (e.g. Takala et al., 2011), or instruments developments (e.g. Schneebeli et al., 1998). Worldwide many study sites have been established for snow monitoring (Ménard et al., 2019). Col de Porte in France (Lejeune et al., 2019), Sodankylä in Finland (Leppänen et al., 2016), and Weissfluhjoch

(WFJ) in Switzerland (Meister, 2009) offer some of the longest time-series of snowpack observations, e.g. dating back to 1936 for the WFJ site. Regular snowpack monitoring programs rely on weekly to bi-weekly manual observations and measurements, digging temporally and spatially consecutive snow pits. Observations comprise mainly traditional profiling with a characterization of layer properties (hand hardness....) and measurements of ram resistance and snow temperatures, all following standard procedures.(Fierz et al., 2009). Those measurements are typically complemented by so-called snow stability test, such as the

compression test (Jamieson, 1999; van Herwijnen and Jamieson, 2007), to monitor weak layers and snow mechanical properties in view of avalanche forecasting. Although these traditional characterization methods are well-established, they suffer from well-known problems of quantitative objectivity, limiting their use for physical snow modeling.

To address this issue, efforts have shown a clear tendency of replacing traditional measurements by newly-developed field methods to obtain more objective, non-empirical snow properties. Concerning the characterization of snow microstructure,

the observer-biased estimate of traditional grain size tends to be replaced by measurements of specific surface area (SSA) (Morin et al., 2013; Leppänen et al., 2015). It is defined by the ice/air interface surface area divided by the snow mass, which is inversely proportional to the optical grain size, and drives many snow processes as metamorphism, radiation interaction, air flow, chemical reactions (e.g. Domine et al., 2007). Different field instruments were developed to measure SSA based on similar methods such as DUFISSS (Gallet et al., 2009), POSSSUM (Arnaud et al., 2011), Iris (Montpetit et al., 2012), or

IceCube (Zuanon, 2013). Concerning snowpack stability assessment, classical stability tests are now often complemented by the propagation saw test (PST), developed about a decade ago to objectively characterize the crack propagation propensity based on the critical crack length parameter (Gauthier and Jamieson, 2006; Sigrist and Schweizer, 2007; van Herwijnen and Jamieson, 2005). The critical crack length corresponds to the length of a saw cut manually introduced in a buried weak layer leading to rapid crack propagation (e.g. Gauthier and Jamieson, 2008). Additional mechanical parameters can be obtained

when combining PSTs with particle tracking velocimetry (van Herwijnen et al., 2016).

These latest advances in field measurements coincide with similar improvements in detailed snowpack models such as Crocus (Brun et al., 1992; Vionnet et al., 2012) and SNOWPACK (Lehning et al., 2002b; Wever et al., 2015). The modeling of SSA as a prognostic variable was included in Crocus to replace the empirical grain size parameter (Carmagnola et al., 2014), and indirectly estimated in SNOWPACK from the grain size, dendricity, and sphericity (Vionnet et al., 2012). Modeling

the SSA allows for an unambiguous comparison with SSA measurements. In addition, many snow properties can now be formulated using physical principles that naturally involve the SSA as a parameter. Likewise, a new model of the critical cut length based on objective stratigraphic information was implemented in SNOWPACK (Gaume et al., 2017) and recently refined to support avalanche risk forecasting (Richter et al., 2019).

These advances, coherently developed in field techniques and modeling, come along with new demands for validation

campaigns. If snow models are only validated against surface or bulk measurements instead of the full stratigraphy, the com-



pensation of effects may prevent the detection of model errors (e.g. Essery et al., 2013; Lafaysse et al., 2017). However, only a few quantitative evaluations of density and SSA profiles exist (Morin et al., 2013; Leppänen et al., 2015; Wever et al., 2015; Essery et al., 2016). Presently, the evaluation of density and SSA is partly limited by the temporal and spatial resolution of measured profiles, which are typically conducted on a weekly to bi-weekly basis with a vertical resolution of 3 cm or set by the layers. In contrast, modeled profiles can be provided hourly and at sub-centimeter vertical resolutions. The gap in resolution between measurements and models precludes the evaluation of snow processes occurring on short time scales and/or locally in the snowpack, such as surface hoar formation (e.g. Stössel et al., 2010), faceting (e.g. Pinzer et al., 2012), or crust formation. Concerning the critical cut length, Richter et al. (2019) reported a good agreement between the temporal evolution of the critical crack length measured in the field and modeled from the refined parameterization. They also highlighted the capability of the parameterization to detect weak layers in simulated snow profiles.

Increasing the spatio-temporal resolution of measurements is still cumbersome due to inherent time-constraints for snow pits and manual measurements. Towards a remedy, recent studies utilized the micro-penetrometer SnowMicroPen (SMP) (Schneebeli et al., 1999) for both, microstructure characterization and stability assessment. Proksch et al. (2015) presented a statistical method to retrieve density and SSA from SMP data and Reuter et al. (2015) suggested an approach to estimate point snow instability from SMP data. These examples exploit key advantages of the SMP, namely fast profiling for frequent measurements and high vertical resolution such as profiles are obtained at a considerably finer scale (mm) than possible with traditional means. Though principally promising, the use of the SMP within snow monitoring programs has never been assessed and would require a comprehensive comparison to other methods to evaluate uncertainties.

In the context raised above, the value of emergent, objective snow properties, their potential to replace traditional means in operational snow monitoring programs, and their requirements on temporal and vertical resolutions for model evaluations can only be investigated within a multi-resolution and multi-instrument dataset to facilitate comprehensive cross-validation analyses. We strive to provide such a resource in the form of the outcome of an extensive snow measurement campaign which is referred to as RHOSSA in resemblance of density ($\rho$) and SSA. The campaign was carried out at the WFJ site from December 2015 to March 2016 and comprises:

- daily (full-depth) profiles of density and SSA of 1 mm vertical resolution derived from SMP measurements

- weekly (full-depth) profiles of density and SSA of 3 cm vertical resolution from manual snow pit measurements

- bi-weekly (full-depth) traditional profiles with layer-dependent vertical resolution, completed with PST and classical stability tests

- occasional (selected locations) profiles of the 3D microstructure at 18 $\mu$m vertical resolution from X-ray tomography.

Our main results comprise (1) a new re-calibration of density and SSA retrievals from SMP 4 measurements, (2) the evolution of density and SSA profiles at unprecedented spatial and temporal resolution, (3) the evolution of snow instability from various stability tests, (4) a comparison of the density and SSA estimates over time for distinct layers of the snowpack, and (5) a comparison between measured values of density and SSA and modeled ones from standard SNOWPACK runs that documents





the state-of-the-art and highlights the potential of high resolution stratigraphy data for snow model evaluation and future developments.

The paper is organized as follows. Section 2 provides an overview of the design of the RHOSSA campaign. Section 3 and Section 4 describe the measurement methods and the simulations with SNOWPACK, respectively. Section 6 presents specific
data analysis methods applied to exploit the RHOSSA dataset, namely a re-defined statistical model for density and SSA retrievals from SMP 4 measurements and a layer tracking method to monitor the evolution of specific layers of the snowpack over the season. Section 6 provides a first analysis of the RHOSSA dataset in terms of stratigraphy, stability, density and SSA, including cross-comparisons between measurements and the evaluation of SNOWPACK simulations. Specific points are finally discussed in Section 7.

## 10  2   Campaign design

During the winter of 2015-2016, the snow observation program at the WFJ site, located in the Eastern Swiss Alps above Davos (elevation of 2536 m, latitude 46.82963 N, longitude 9.80925 E), was supplemented with additional measurements, forming all together the RHOSSA field campaign. We focused on the period of dry snow from beginning of December 2015 to end of March 2016, to ensure measurements in dry snow condition as required by some of the used instruments. In addition,
measurements were done in the morning typically starting at 8am. The RHOSSA campaign included traditional profiling, stability tests, density cutter measurements, IceCube measurements, SMP measurements, and tomography. Using such a wide range of measurement methods resulted in different temporal resolutions (frequency) and spatial resolutions (vertical along the snow profile), as synthesized in Table 1. SMP measurements were performed daily, density cutter measurements and IceCube measurements were performed once a week, traditional snow profiles were recorded on a weekly to bi-weekly basis
and completed with stability tests. X-ray tomography measurements of extracted, decimeter-sized samples were occasionally performed six times during the season. Spatial resolutions range from 0.1 mm for the tomography-based properties to the size of the snow layer for the traditional profiling (typically from 1 to 30 cm).

The measurement field at the WFJ site is a flat area of about $20 \times 8$ m$^2$ (Fig. 1). To ensure an efficient use of the snow field, measurements were performed within defined areas. The snow field was divided into three corridors, each 20 m long
and 1.5 m wide, as illustrated in Figure 1. Throughout the season, sets of measurements were performed moving continuously along the corridor in daily steps, starting at one end of corridor 1 and ending at the end of corridor 3, two consecutive sets of measurements being at least 30 cm apart to avoid disturbances. A schematic of the location of three consecutive sets of measurements ("day 1", "day 2", and "day 3") performed in corridor 2 at mid season is shown in Figure 1. Each corridor was divided lengthwise in 2 parts of 75 cm wide. One side was reserved for stability tests (red area in Fig. 1); the other side was used
for all the other measurements. First, the five daily SMP measurements with a 15 cm spacing were performed perpendicular to the corridor direction (black dots in Fig. 1). Then, during a snow pit day as illustrated by "day 2" in Figure 1, the pit was dug such that the pit wall was parallel and a few centimeters behind the line that was formed by the SMP measurements. Density cutter and IceCube measurements were done next to each other (blue and orange areas in Fig. 1), and complemented by a





traditional snow profile when needed (green area in Fig. 1). Finally, for the occasional X-ray tomography, undisturbed snow blocks were extracted from the pit wall near the location of the other measurements.

**Table 1.** Overview of the RHOSSA campaign measurements.

| Method | Frequency | Vertical resolution | Measured or derived properties |
|---|---|---|---|
| SnowMicroPen | daily | 1 mm | penetration force, density, SSA |
| Density cutter | weekly | 30 mm | density |
| IceCube | weekly | 30 mm | SSA |
| Traditional profile | every 1 to 2 weeks | variable | traditional layer parameters, temperature, hand hardness, ram resistance |
| Stability tests | 8 times over the season | - | critical crack length, #taps until failure |
| Tomography | 6 times over the season | 0.1 mm | density, SSA |

## 3 Measurements

### 3.1 Traditional profile and stability tests

Traditional snow profiles were observed to characterize snow stratigraphy by hand hardness, grain size and grain type. In addition, ram resistance, snow temperatures, and water equivalent of the snow cover were measured (Fierz et al., 2009). Snow stability tests were performed to identify potential weak layers and evaluate the load required for failure. Specifically, we performed the compression test (CT; van Herwijnen and Jamieson, 2007), the extended compression test (ECT; Simenhois and Birkeland, 2009) and the propagation saw test (PST; Gauthier and Jamieson, 2008). In a CT or an ECT, the snowpack is

progressively loaded by tapping on a snow shovel placed on the snow surface with increasing force (10 taps from the wrist, 10 taps from the elbow and 10 taps from the shoulder). If a failure occurs within the snow cover, the loading step, i.e. the number of taps at which the failure occurred, is recorded. In a CT, which consists of an isolated column of 30 by 30 cm, information describing the type of failure is also recorded (for more details see van Herwijnen and Jamieson, 2007). In an ECT, which consists of an isolated column of 30 by 90 cm, the propagation distance across the column is recorded as either no propagation,

partial propagation or full propagation (for more details see Simenhois and Birkeland, 2009). CT and ECT are thus used to identify potential weak layers and qualify the loading required for failure. The PST, on the other hand, is used to measure the critical crack length required for crack propagation in an a priori known weak layer. It consists of an isolated 30 cm wide



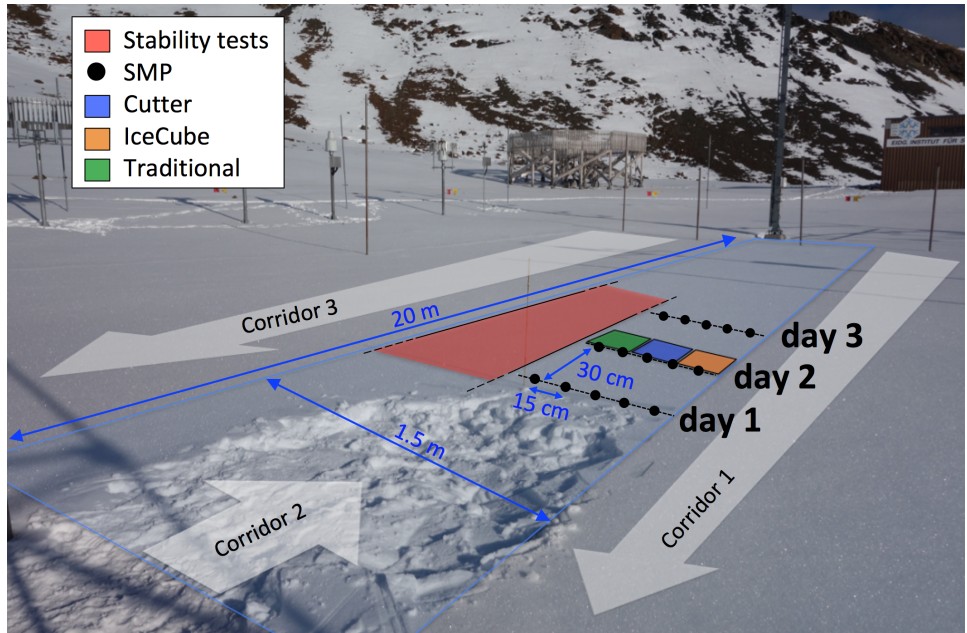

**Figure 1.** Picture of the snow field where measurements of the RHOSSA campaign were performed. The location of each measurement is illustrated for three consecutive days.

column with a length of at least 120 cm, which has been excavated to below the weak layer of interest. An artificial crack is then created by drawing a snow saw through the weak layer until the critical crack length is reached and rapid crack propagation occurs. The critical crack length is recorded as well as the propagation distance, where END refers to cracks which propagated to the end of the column (for more details see Gauthier and Jamieson, 2008).

## 3.2 Density cutter

A density cutter was used to manually record the density profile of the snowpack by performing successive measurements from the surface to the bottom of the snowpack with a vertical resolution of 3 cm. A box-type density cutter of 100 cm$^3$ ($3 \times 5.5 \times 6$ cm) (Carroll, 1977; Conger and McClung, 2009; Proksch et al., 2016), was used to measure density by weighing a snow sample extracted from the cutter. A measurement error of about 10% can be expected (Carroll, 1977; Conger and McClung, 2009; Proksch et al., 2016), typical source of errors being the measurement of compacted snow volumes (overestimation) when extracting light snow, and of incomplete snow volumes (underestimation) when extracting fragile snow (e.g. faceted crystals or depth hoar).

## 3.3 IceCube

The IceCube was used to measure an SSA profile of the snowpack by performing successive IceCube measurements from the surface to the bottom with a vertical resolution of 3 cm. The IceCube is an optical system commercialized by A2 Photonic Sen-




sors (Zuanon, 2013) to retrieve SSA from measurements of the infrared hemispherical reflectance of snow (Gallet et al., 2009). Briefly, a snow sample is illuminated with a 1310 nm light diode and the light reflected by the snow surface is recorded. The signal is recorded as voltage values then converted in reflectance values based on a voltage-to-reflectance calibration curve obtained using certified optic standards. SSA values are finally estimated from the reflectance values using the parametrization of Gallet et al. (2009). The complete description of the measurement principle can be found in Gallet et al. (2009). Measurements were performed on cylindrical snow samples with a 6 cm diameter and 2.5 cm height, extracted from the snow pit following the method given by Gallet et al. (2009) and Zuanon (2013). Measurement uncertainty was estimated to about 10% for SSA values below 60 m$^2$ kg$^{-1}$. Additional measurement artifacts occur for snow with higher SSA that can lead to over-estimated SSA values (Gallet et al., 2009).

## 3.4 SnowMicroPen

The SnowMicroPen (SMP), a digital cone penetrometer, was used to measure the vertical penetration resistance profile of the snowpack. From that, density and SSA profiles were derived based on a statistical model and after a specific signal processing, as described in Section 5.1. The SMP consists of a motorized probe that is driven vertically into the snowpack at a constant speed of 20 mm s$^{-1}$ to measure the penetration resistance exerted on a cone (diameter of 5 cm and cone half angle of 30°) located at the tip of the probe (Schneebeli et al., 1999). We used a version 4 SMP with a 2-meter rod and recorded penetration resistance with a vertical resolution of 1/242 mm. Two preliminary measurements were systematically performed to cool the SMP towards snow temperature before the five daily measurements were taken. The quality of each SMP profile was manually checked by evaluating the penetration resistance profiles. Signals showing strong drifts were discarded (e.g. frozen water in the SMP motor, defect of the force sensor, etc). Signals that correspond to measurements in the air and in the ground were truncated. No offset correction was necessary for this dataset.

## 3.5 Micro-computed tomography

X-ray micro-computed tomography was used to image the 3D microstructure of snow samples extracted from the snowpack at selected locations. Snow blocks of about $30 \times 30 \times 30$ cm$^3$ were cut out from the profile wall on 14 December, 13 January, 27 January, 10 February, 16 February, and 2 March. The location of the extracted blocks within the snowpack were chosen subjectively, either to ensure temporal continuity with a previously sampled block, or to re-focus on a particular layer of interest, mainly persistent weak layers. Extracted blocks were sealed in Styrofoam boxes and filled with dry ice (about -80°C) for transportation from the field site to the cold lab (duration approximately 1 h). In the lab, the blocks were stored at -25°C, and successively sub-sampled into sample holders of 7 cm height and 3.6 cm diameter. These samples were then scanned in a cooled micro-computer tomograph ($\mu$CT 80, Scanco Medical) with a resolution of 18 $\mu$m voxel size. Reconstruction followed standard procedure. The reconstruction utilized standard procedures with noise reduction by Gaussian filtering (support=2 voxels, width=1.2 voxels) and binary segmentation following the method of Hagenmuller et al. (2013). From the binary 3D images, density and SSA were computed over a moving window of 120 pixels height obtaining profiles at a vertical resolution of about 2 mm.





## 4 Simulations with SNOWPACK

To put the measurement campaign in context, we conducted standard simulations with the detailed snow cover model SNOW-PACK (Lehning et al., 2002b) using version 3.4.1, revision 1473 (https://models.slf.ch/p/snowpack/). SNOWPACK model was driven with an optimized dataset of meteorological and snowpack measurements from the automatic weather station at the WFJ

site (WSL Institute for Snow and Avalanche Research SLF, 2015). The dataset contains standard meteorological measurements including air temperature (ventilated), relative humidity (ventilated), wind speed, shortwave and down-welling long wave radiation. The snow cover mass balance was driven with the increments of measured snow depth. To estimate the occurrence of rainfall events if air temperature exceeded 1.2 °C, data on precipitation was used additionally. Snow albedo was forced from the in-situ measurements of incoming and reflected shortwave radiation fluxes and snow height. The calculated values

underwent a plausibility check and in case of a negative outcome were replaced by the model parametrization. The surface sensible and latent heat flux parameterizations are derived from Monin–Obukhov similarity (Lehning et al., 2002a). Neumann boundary conditions were used at the snow-atmosphere boundary whereas a constant geothermal heat flux (0.06 W m$^{-2}$) was assumed at the bottom of the 3 m deep soil column. Liquid water flow in snow was solved using Richards equation recently implemented by Wever et al. (2014). The time step for the simulation was 15 min and output was written every 60 min. For this

campaign, we were particularly interested in evaluating the model in terms of density and SSA. The density of new snow was obtained from an empirical relation between air temperature and wind speed (Schmucki et al., 2014). The snowpack itself is considered to be a linear viscoelastic material, the settlement of which was calculated as described in section 2.2.2 in Lehning et al. (2002b), using an altered viscosity parametrization. In addition, the effect of load rate was taken into account but any elastic effects were neglected. SSA was simply retrieved from the optical diameter of snow that is empirically derived from

dendricity, sphericity, and grain size according to Vionnet et al. (2012).

## 5 Data analysis methods

### 5.1 Deriving density and SSA from SMP

As a prerequisite to derive density and SSA from SMP measurements, it was necessary to modify the current statistical models of Proksch et al. (2015). When applying the parametrizations of Proksch et al. (2015), SMP-derived density and SSA compared

rather poorly to values from cutter and IceCube measurements respectively (Fig. 2). This is in part due to the fact that the parametrizations of Proksch et al. (2015) were derived from measurements with an SMP device version 2 whereas we used a newer SMP version 4 that contains different electronic components leading to different force correlations at small scale. We thus derived a re-calibration of the statistical models of Proksch et al. (2015) to better match our snow pit measurements. The obtained density and SSA parametrizations are called new parameterizations hereafter.

The idea of Proksch et al. (2015) was to relate some relevant SMP micro-parameters to reference values of density (or SSA), both obtained from independent, co-located and co-temporal measurements, using a statistical regression model. Here we followed the same procedure but we took our cutter measurements as reference values of density ($\rho_{\text{cutter}}$) and our IceCube





measurements as reference values of SSA (SSA$_{ic}$), whereas Proksch et al. (2015) used values from tomography measurements. The statistical modeling was thus applied based on a sub-dataset of 15 days where both SMP and snow pit measurements were available. The SMP micro-parameters consist of the median of the penetration resistance force $\tilde{F}$ and a characteristic length of the microstructure $L$ (akin to the distance between two ruptures), as defined in the stochastic model of Löwe and van

Herwijnen (2012). Both parameters were computed from the raw penetration force profiles over a sliding window of 1 mm with 50% overlap, yielding profiles of $\tilde{F}$ and $L$ with a vertical resolution of 0.5 mm. Note that Proksch et al. (2015) used a sliding window of 2.5 mm, but tests with different window heights (1, 2.5 and 5 mm) did not show a significant impact. A median operation was applied to the five profiles of $\tilde{F}$ and $L$ obtained per day to get one representative profile per day; the latter was then averaged vertically using a 3 cm window to match the vertical resolution of the cutter and IceCube measurements. Finally,

profiles of $\tilde{F}$, $L$, $\rho_{cutter}$ and SSA$_{ic}$ were aligned by simply using snow surface as common reference and cropped to the length of the shortest profile. Based on this sub-dataset, we applied a regression of the form

$$\rho_{smp} = a_1 + a_2 \ln(\tilde{F}) + a_3 \ln(\tilde{F}) L + a_4 L \tag{1}$$

to estimate density from $\tilde{F}$ and $L$ by least-squares optimization ($\rho_{cutter}$ being the target). The following parameters were obtained: $a_1 = 295.8 \pm 0.3$, $a_2 = 65.1 \pm 0.1$, $a_3 = -43.2 \pm 0.4$, and $a_4 = 47.1 \pm 0.7$, where $\rho_{smp}$ is in kg m$^{-3}$, $L$ in mm and

$\tilde{F}$ in N. This regression has a R$^2$ coefficient of 0.79, a residual standard error of 40.8 kg m$^{-3}$, and p-values less than $10^{-3}$. Differing slightly from the one suggested by Proksch et al. (2015), a regression of the form

$$SSA_{smp} = b_1 + b_2 \ln(L) + b_3 \ln(\tilde{F}) \tag{2}$$

was applied to estimate SSA by least squares optimization (SSA$_{ic}$ being the target). The following regression parameters were obtained: $b_1 = 0.57 \pm 0.05$, $b_2 = -18.56 \pm 0.04$, and $b_3 = -3.66 \pm 0.01$, where SSA$_{smp}$ is in m$^2$ kg$^{-1}$. This regression has

a R$^2$ coefficient of 0.67, a residual standard error of 8.4 m$^2$ kg$^{-1}$, and p-values less than $10^{-3}$.

The performance of the present parametrizations (Equations 1) and (2) compared to the original parametrizations (Proksch et al., 2015) is shown with observed density from cutter measurements and observed SSA from IceCube measurements in Figure 2. Note that these scatter plots shows values from the same sub-dataset used for the statistical analysis above but profiles were re-aligned using the height of a thin persistent well-defined layer (described in Sec. 6) instead of the snow

surface, leading to a better vertical match of the profiles and thus a better correlation between estimates from SMP and snow pit measurements. As expected, SMP-derived properties are closer to the snow pit measurements when using the present parametrizations. Between $\rho_{cutter}$ and $\rho_{smp}$, a R$^2$ coefficient of 0.84 is found when using Eq. (1) against 0.73 when using the parametrization of Proksch et al. (2015). Between SSA$_{ic}$ and SSA$_{smp}$, a R$^2$ coefficient of 0.81 is found when using Eq. (2) against 0.64 when using the parametrization of Proksch et al. (2015). Hence, the present parametrizations Eq. ((1)) and ((2))

were applied to retrieve density and SSA from the entire SMP data.

## 5.2 Layer tracking

We present a method to track particular layers of the snowpack throughout the season and retrieve their properties. This method allows evaluating measurement methods and/or simulation results by comparing the properties of the tracked layers, as





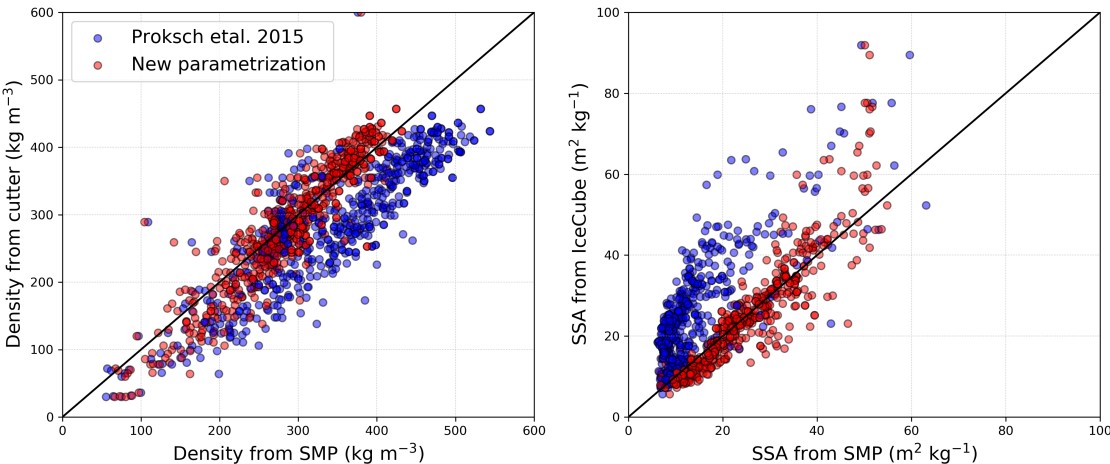

**Figure 2.** Left: Density from cutter measurements against density derived from SMP data using the parametrization of Proksch et al. (2015) (blue circles) and the re-calibrated present parametrization (red circles). Right: SSA from IceCube measurements against SSA derived from SMP data using both parametrizations.

presented in Section 6.2 and 6.3. To do so, the layers of interest were defined by a upper and lower boundary, a boundary being detected by a significant, often sharp transition in the vertical profile of snow properties (either density, SSA, or penetration force). Boundaries were manually identified for all measurement methods by simply looking at the property profiles (density, SSA, or penetration resistance) and reporting heights. For the SMP data, this step was performed on the median profiles of

penetration resistance force computed from the five daily SMP measurements. The identification of layer boundaries was sometimes challenging for weak stratigraphic transitions, e.g. the transition between a layer of fresh snow that fell onto a soft snow layer. To this end, boundaries were backtracked in time, starting from a profile where the layer is older (typically 1 month after its deposition) and its boundaries more clearly detectable. When approaching the date of the layer deposition, additional information, such as observed height of new snow, was sometime used to help delineate boundaries. Once boundaries of the

layers of interest were defined on all measurements of our dataset, layer properties were computed by averaging data within heights given by the referenced boundaries.

We used a different method to identify layers in the SNOWPACK simulations based on the layer deposition date that is one of the layer properties. To do so, we attributed a time stamp (YYMMDD) to each layer boundary that corresponds to the date of deposition of the adjacent layer above the given boundary (date of burial). Time stamps were determined using automatic

weather station data as well as the daily manual observations of the snow surface. A layer of interest was then simply defined as simulated layers with a deposition date older than the time stamp of its lower boundary but younger than the time stamp of its upper boundary.



Four distinct layers were tracked in this study and consist in four directly adjacent layers located in the bottom part of the snowpack. We choose these layers because they are among the main stratigraphic features of the snowpack observed during the winter, showed a wide range of snow types and properties, could be tracked over the entire winter, and were relatively easy to identify (rather sharp property transitions). These tracked layers are called the DH-layer (depth hoar), the

MF-layer (melt forms), the FC-layer (faceted crystals), and the RG-layer (rounded grains), from bottom to top layers, referring to the predominant grain shape observed in the layer. They are described in details in the next section. These four layers were identified based on four boundaries called 151201-boundary, 151202-boundary, 160102-boundary, and 160117-boundary, from the lower to the upper boundary, and the ground. This way, the DH-layer was comprised between the ground and the 151201-boundary, MF-layer between the 151201-boundary and the 151202-boundary, FC-layer between the 151202-boundary and the

160102-boundary, and RG-layer between the 160102-boundary and the 160117-boundary.

## 6    Dataset analysis

This section presents a basic analysis of the RHOSSA campaign alongside with measurement inter-comparisons and a preliminary evaluation of the SNOWPACK simulations. To compensate for the inevitable height mismatches of the vertical property profiles, inherent to the snow spatial variability and measurements variability (e.g. Hagenmuller and Pilloix, 2016), all the

profiles presented in the following were re-aligned such as $z = 0$ cm corresponds to the height of the upper boundary of the MF-layer (i.e. the 151202-boundary).

### 6.1    Evolution of weather, snow stratigraphy and stability

To provide background information for the origin of stratigraphic features of the season, Figure 3 shows the seasonal evolution of air and snow surface temperature as well as total snow height and height of new snow over 24 hours. The bi-weekly

traditional profiles observed between 14 December 2015 and 15 March 2016 are presented in the upper caption of Figure 4. We can first note that winter 2015-2016 showed a below-average snow height, especially at the beginning of the season (Fig. 3). End of November, the winter started with a precipitation event after which the snow height reached approximately 40 cm. Thereafter, a dry period followed during which snow surface temperature remained between -20°C and -10°C, allowing large temperature gradients to build up across the shallow snowpack. Traditional profiles show that this basal layer recrystallized

predominantly into depth hoar (dark blue colored layers below 0 cm in Fig. 4, upper panel), although faceted crystals and melt forms were sometimes also reported (light blue and red colored layers), and persisted throughout the season. This basal layer corresponds to the tracked layer referred as the DH-layer (Sec. 5.2). On the late afternoon of 1 December 2015, observers from the nearby ski resort reported rainfall up to 2600 m, and measured snow surface temperature reaching 0°C while the air temperature remained colder (see inset in Fig. 3) indicating freezing rain. This rainfall event led to the formation of a melt-

freeze crust / rain crust at the snow surface, as reported in the traditional profile that followed on 14 December (Fig. 4, red and turquoise colored layer at 0 cm). This crust was persistent throughout the season and tracked as the MF-layer. Mid-December, about 10 cm of new snow accumulated on this crust and recrystallized into faceted crystals by the end of December, favored



by a period of rather clear weather leading to low snow surface temperatures (Fig. 3). Again, this layer of faceted crystals was observed throughout the season (light blue colored layers between about 0 cm and 10 cm in Fig. 4, upper) and corresponds to the tracked FC-layer. January was generally characterized by a more cloudy weather with consistent precipitation events (Fig. 3). With the first snow falls early January, snow accumulated on top of the FC-layer and was quickly buried by the

subsequent heavy precipitation events, being buried under around 75 cm of snow by mid-January. This layer was protected from significant temperature gradients and evolved into small faceted crystals and rounded grains (light blue and light red colored layers between about 10 to 25 cm in Fig. 4). As this layer showed systematically a higher hand hardness (4 fingers against 1 finger) and a smaller grain size (not shown) than the FC-layer and DH-layer, this layer was named RG-layer for a sake of differentiation. Finally, after further precipitation events mostly occurring early February and early March, the snowpack

height reached about 200 cm by mid March and consisted mostly of layers of rounded grains on a weaker base of facets and depth hoar.

   The snowpack stratigraphy simulated by SNOWPACK is shown in the lower panel of Figure 4. Qualitatively, modeled stratigraphy compared well with observed stratigraphy. Indeed, although many subtle differences in grain shape and hand hardness exist throughout the season, the major stratigraphic features are well reproduced, notably the weak base layers (DH-

layer and FC-layer) as well as the overlying slab which mostly consisted of small rounded or faceted grains for which the hardness increases from top to bottom. One major discrepancy is that the melt-freeze / rain crust which formed on 1 December (MF-layer) was not simulated by SNOWPACK (see dedicated comment in Sec. 7.3). Instead, SNOWPACK simulated around 3 cm of new snow, which later re-crystallized into faceted crystals.

   Snow stability tests showed that the weak base, namely the DH-layer and FC-layer, were the most critical weak layers during

most of the season. As shown in Figure 5, both layers consistently failed in CT and ETC until the beginning of February. Thereafter, these layers were not reactive anymore as tapping on the snow surface was not affecting the weak base buried below the hard and thick slab (black symbols in Fig. 5). From the PST, it was possible to follow the evolution of the critical crack length throughout the season (crosses in Fig. 5). Overall, the critical crack length increased steadily from about 20 cm in mid-January to around 60 cm beginning of March for both FC-layer and DH-layer, indicating weak layers less and less

prone to crack propagation with time. Note that the critical crack length was consistently lower for the DH-layer than for the FC-layer.

## 6.2   Evolution of density

Figure 6 presents the evolution of the density profile during the course of the winter, as recorded from density cutter measurements, derived from SMP measurements, and simulated by SNOWPACK. Boundaries of the tracked layers are identified with

solid black lines. The snowpack evolution is characterized by the punctual presence of new snow at the surface, showing the lowest density values down to about 50 kg m$^{-3}$. Overall, snow gets gradually denser upon deeper burial in the snowpack and as the season progresses, reaching density values as high as 450 kg m$^{-3}$ in the middle of the snowpack by mid-winter. Despite located in the bottom of the snowpack, the persistent weak layers (DH-layer and FC-layer) remain significantly lighter than the adjacent layers. Finally, density of the MF-layer remains roughly constant throughout the winter at around 350 kg m$^{-3}$.



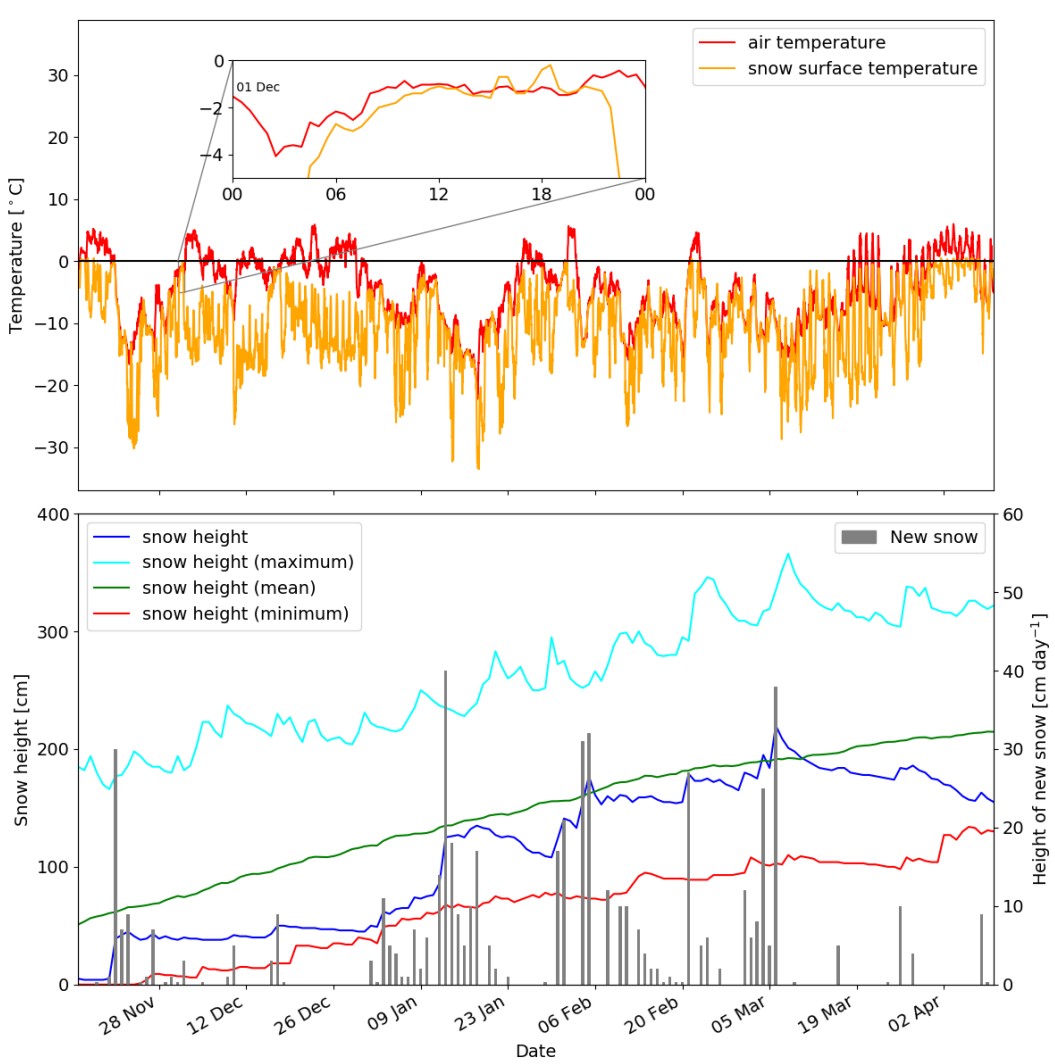

**Figure 3.** Top: Evolution of air temperature (red) and snow surface temperature (orange) at the WFJ site during winter 2015-2016. The inset shows data recorded on 1 December 2015 when the MF-layer formed. Bottom: Seasonal evolution of snow height (blue) and height of new snow (gray bars). For context, the 80 year daily maximum (cyan), minimum (red) and mean (green) snow height are also shown.



**Figure 4.** (top) Manual snow profiles observed during the 2015-2016 winter season. The colors indicate the major grain shape (red: melt forms, light blue: faceted crystals, blue: depth hoar, pink: rounded grains, green: decomposing and fragmented particles, light green: precipitation particles) and the width indicates the hand hardness. Snow height is relative to the top of the MF-layer. (bottom) Simulated snow profiles for the same dates.

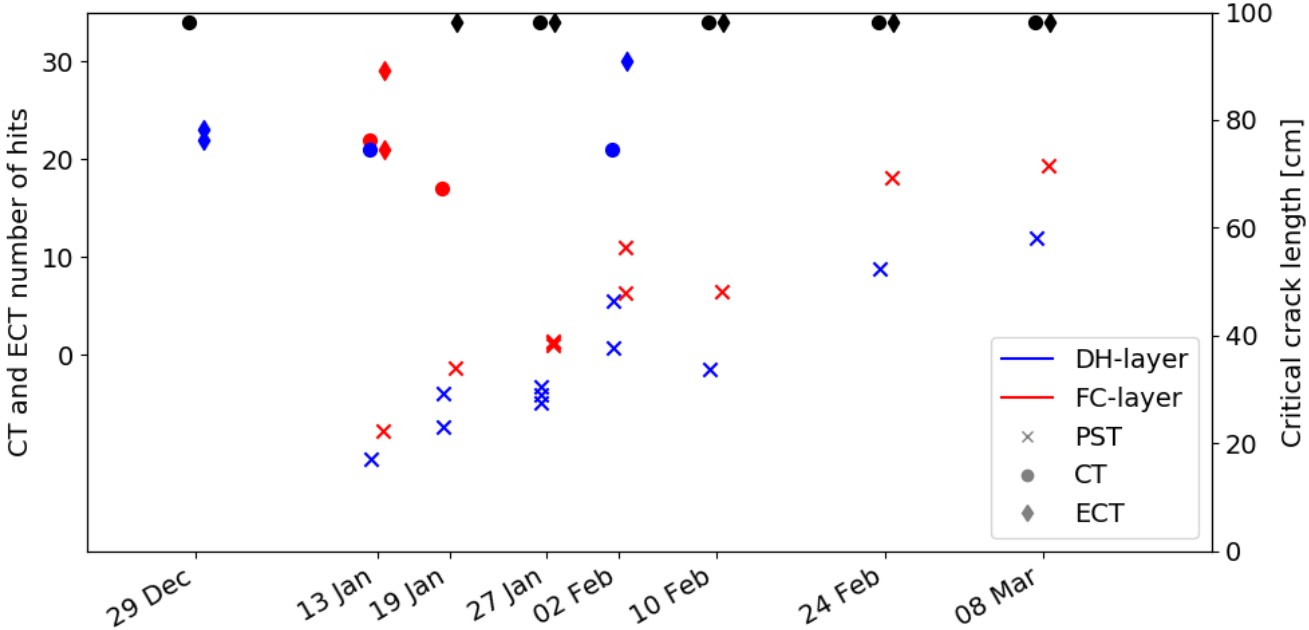

**Figure 5.** Stability tests results for the DH-layer (blue) and FC-layer (red). The number of hits for CT (circles) and ECT (diamonds) and the critical crack length obtained from the PST (crosses) are shown. Black symbols indicate that the CT or the ECT did not result in a failure in the layers.

Although these features are consistently reported by both measurement methods, many stratigraphic details are only revealed by the SMP measurements and are not captured by the cutter measurements. The high temporal and spatial resolution of the SMP measurements allows indeed following almost continuously the evolution of density with time. For instance, we can clearly follow the density evolution of the 2 cm thick snow layer from its formation on February 22 showing density values
around 350 kg m$^{-3}$ (layer located at 145 cm height on February 22 in Fig. 6b) to mid-March when buried under about 40 cm of snow but still showing similar density values (layer located at 115 cm height on March 15 in Fig. 6b). The evolution of this layer is not or only diffusely captured by the cutter measurements. Note that this layer was reported in the traditional profiles from the 24th of February on as a layer of melt forms with a hand harness of one fist (Fig. 4).

Allowing further comparisons, Figure 7 provides a comparison of the vertical profiles of density on January 13, 2016 and
March 2, 2016. Figure 8 shows the evolution of density for the 4 tracked layers DH-layer, MF-layer, FC-layer, and RG-layer throughout the winter. Both figures highlight an overall consistency between measurements. A slightly larger scatter is observed in the density evolution of the MF-layer (Fig. 8b), which might be partly due to uncertainties in the definition of the layer boundary (see Sec. 7.1). One can also note the decrease in density recorded by the last two cutter measurements for the DH-layer and FC-layer (Fig. 8a and 8c). This might reflect a measurement bias that can occur when sampling fragile snow
layers (under-sampling).





Simulations of the density profiles over the season agree overall well with the observations (Fig. 6c). The mis-modeling of the MF-layer, as mentioned earlier, leads however to large local deviations. Moreover, SNOWPACK seems to overestimate the densification rate of the DH-layer and FC-layer, leading to significantly higher modeled values by mid-March (Fig. 8a and 8c). This overestimation can also be observed in the vertical profile of March for both weak layers (Fig. 7b). Inversely,

densification rate seems to be underestimated for layers evolving from fresh snow to rounded grains in the upper part of the snowpack, leading to simulated densities lower than the measured ones by mid-march, as shown in Figure 6 and 7b (layers from about 20 to 100 cm height). Finally, other inconsistencies can be observed locally in the simulated stratigraphy, such as the two relatively denser layers observed near the surface in March 2 at around 125 cm and 135 cm (Fig. 7b).

### 6.3  Evolution of SSA

Figure 9 shows the evolution of the SSA profiles over the course of the winter from IceCube measurements, from SMP measurements, and from SNOWPACK simulations. Note that IceCube measurements could not be performed on 19 January 2016 and 10 February 2016. SSA values range from about 70 $m^2 kg^{-1}$, for fresh snow layers at the surface, to about 5 $m^2$ $kg^{-1}$, in the bottom part of the snowpack. The MF-layer, well identifiable in terms of density (Fig. 6a and b), is here difficult to distinguish from the DH-layer and the FC-layer due to their similar SSA values. The general trend of the SSA evolution is

an overall decrease with time and depth. The impact of the spatial and temporal resolution is again highlighted. For instance, the evolution of the layer deposited on February 22, easily identified by lower SSA values (greenish colors) than the ones of the adjacent layers, is clearly captured by the SMP measurements but only diffusely reported in the IceCube data.

To compare further, the vertical profiles of SSA of 13 January 2016 and 2 March 2016 are shown in Figure 10, and the temporal evolution of the SSA of the 4 distinct layers (DH-layer, MF-layer, FC-layer and RG-layer) is presented in Figure 11.

In particular, the latter figure allows analyzing the SSA decrease with time. The RG-layer shows the largest decrease, especially shortly after deposition when SSA evolves from about 45 to 20 $m^2 kg^{-1}$ within one week. The SSA decay in the MF-layer and the DH-layer is slower, decreasing from about 15 to 10 $m^2 kg^{-1}$ within the whole course of the season.

Both figures highlight significant disagreements between measurement methods. To further investigate this issue, Figure 12 presents vertical profiles of SSA for all the five dates when tomography, IceCube and SMP measurements were performed. In

all profiles, SSA values from IceCube measurements are systematically higher than values from tomography measurements, by a factor of about 1.3. Besides this systematic bias, large deviations are found on 13 January 2016 in the upper half of the snowpack, for which SSA values from IceCube measurements range from 60 and 100 $m^2 kg^{-1}$, whereas values from SMP measurements do not exceed 50 $m^2 kg^{-1}$ (Fig. 12b, upper 60 cm). Possible causes for these deviations are discussed in Section 7.3.

Finally, SNOWPACK overall underestimates SSA compared to measurements (Fig. 9, 10, and 11). Deviations are higher with the IceCube data than the tomographic data, for which some good agreements can locally be found, for instance when looking at the SSA evolution of the tracked layers from mid-January on (excluding the MF-layer).



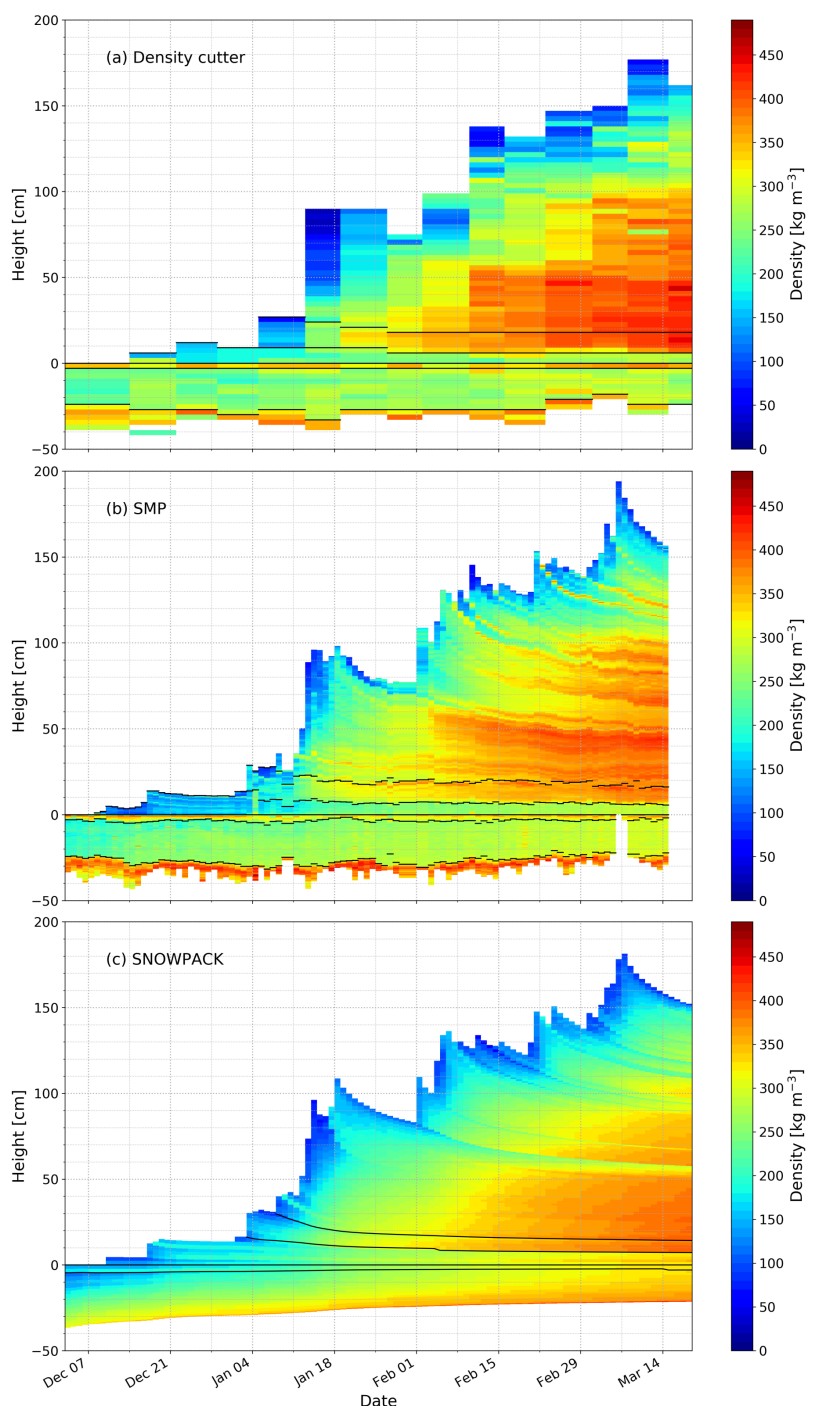

**Figure 6.** Evolution of the density profile during winter 2015-2016 (a) from cutter measurements, (b) derived from SMP measurements, and (c) simulated by SNOWPACK. Boundaries shown with black lines allow identifying the 4 tracked layers (DF-layer, MF-layer, FC-layer and RG-layer, from bottom to top). Measurements below the lowest boundary shown in SMP and cutter data were not considered part of DF-layer.

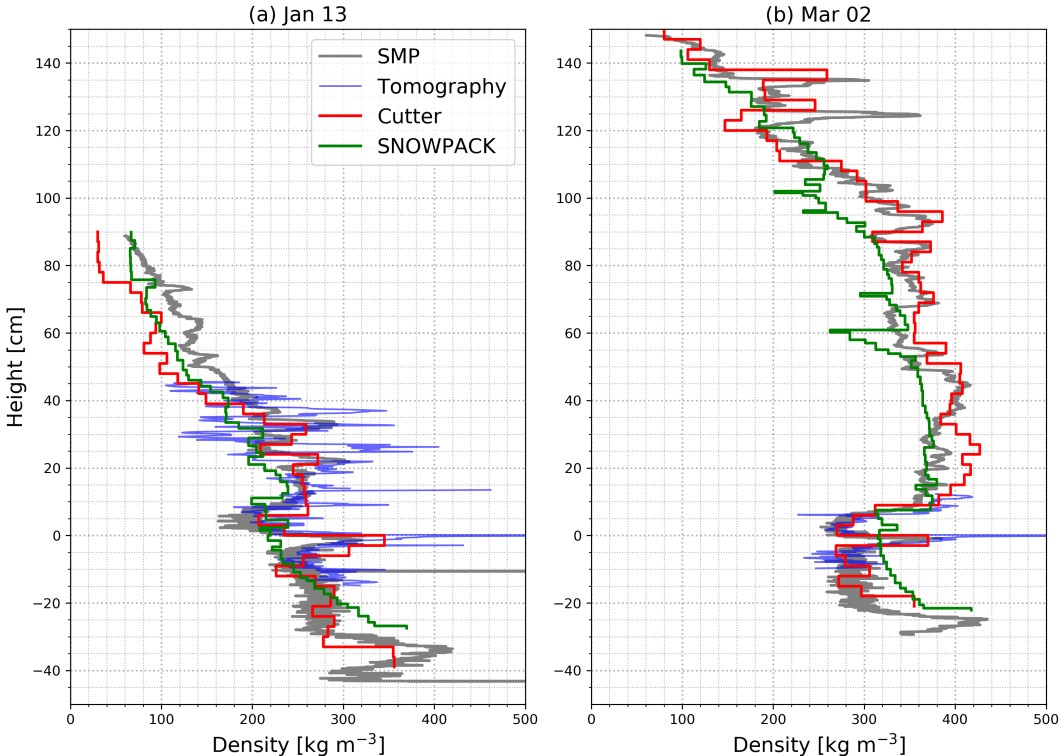

**Figure 7.** Vertical profile of density on 13 January 2016 and 02 March 2016 from SMP, cutter, and tomography measurements as well as modeled by SNOWPACK.

## 7 Discussion

### 7.1 The RHOSSA dataset for snow model evaluation

The presented dataset can be utilized as validation data for the evaluation of snow model outputs for the case of a dry alpine snowpack and over one winter season. Output parameters that can be evaluated are density, SSA, critical cut length, traditional snow pit measurements (grain size, grain type, hardness, temperature) and results from compression and extended compression tests. Snow models can be driven using the optimized forcing dataset, which includes meteorological and snow data from automatic and manual observations, provided in this study (Sec. 4). The RHOSSA dataset alone does not allow for robust and complete model evaluations, as model performances can vary depending on years and sites (Essery et al., 2013; Krinner et al., 2018). Yet, the snowpack monitored over winter 2015-2016 offered a wide range of snow type and property variations throughout the season. It included typical persistent weak layers at the bottom of the snowpack (DH-layer and FC-layer) relevant for stability assessment for avalanche risk forecasting. Although the study focused on dry snowpack, some rain/melt events are also represented by the presence of several melt-refreeze crusts.





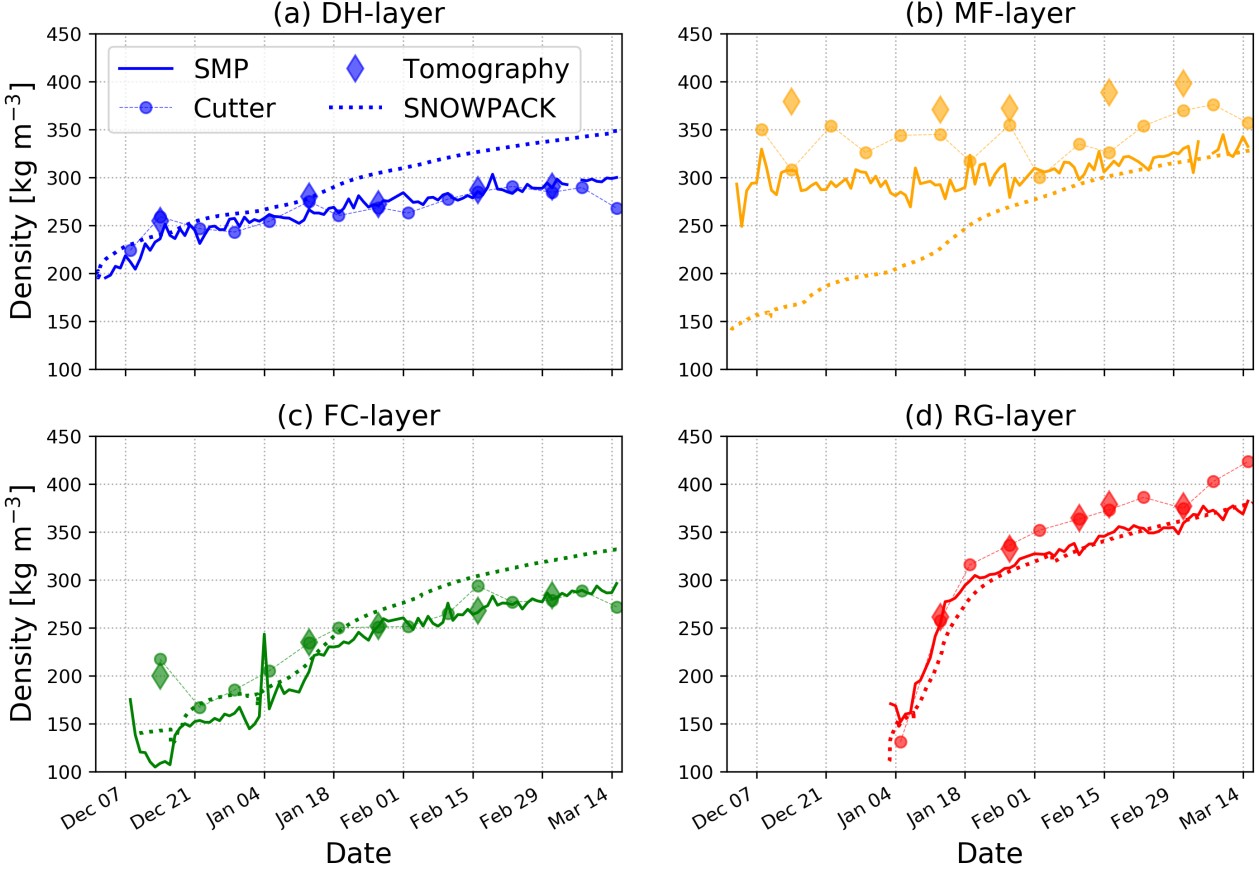

**Figure 8.** Density evolution of the 4 tracked layers from SMP, density cutter, and tomography measurements as well as modeled by SNOW-PACK.

The specificity of the RHOSSA dataset is to provide time-series of density and SSA at a daily frequency and with a vertical resolution of 1 mm, in contrast with previous validation datasets (weekly to bi-weekly, vertical resolution of 3 cm or higher) (e.g. Morin et al., 2013; Leppänen et al., 2015). Both temporal and spatial resolution are critical to account for in snow models because thin layers as well as processes occurring within short-time scales can have a significant impact on the snowpack behavior, e.g. on its mechanical stability (e.g Jamieson and Johnston, 1992). We highlight the need of high resolution datasets, as provided here, to evaluate the simulation of such features and processes.

In addition to validation datasets, comparison methods are also crucial when assessing models. Different methods were presented in the past to compare measurements and simulations: i) the comparison of averaged (bulk) values over the entire snowpack height (e.g. Landry et al., 2014; Leppänen et al., 2015; Essery et al., 2016), which is easy to implement but provides rather limited information, ii) the comparison of paired-values at the same height of the snowpack, which allows assessing the snowpack stratigraphy (e.g. Lehning et al., 2001; Morin et al., 2013) (as in Fig. 7 and 10), and iii) the comparison of values



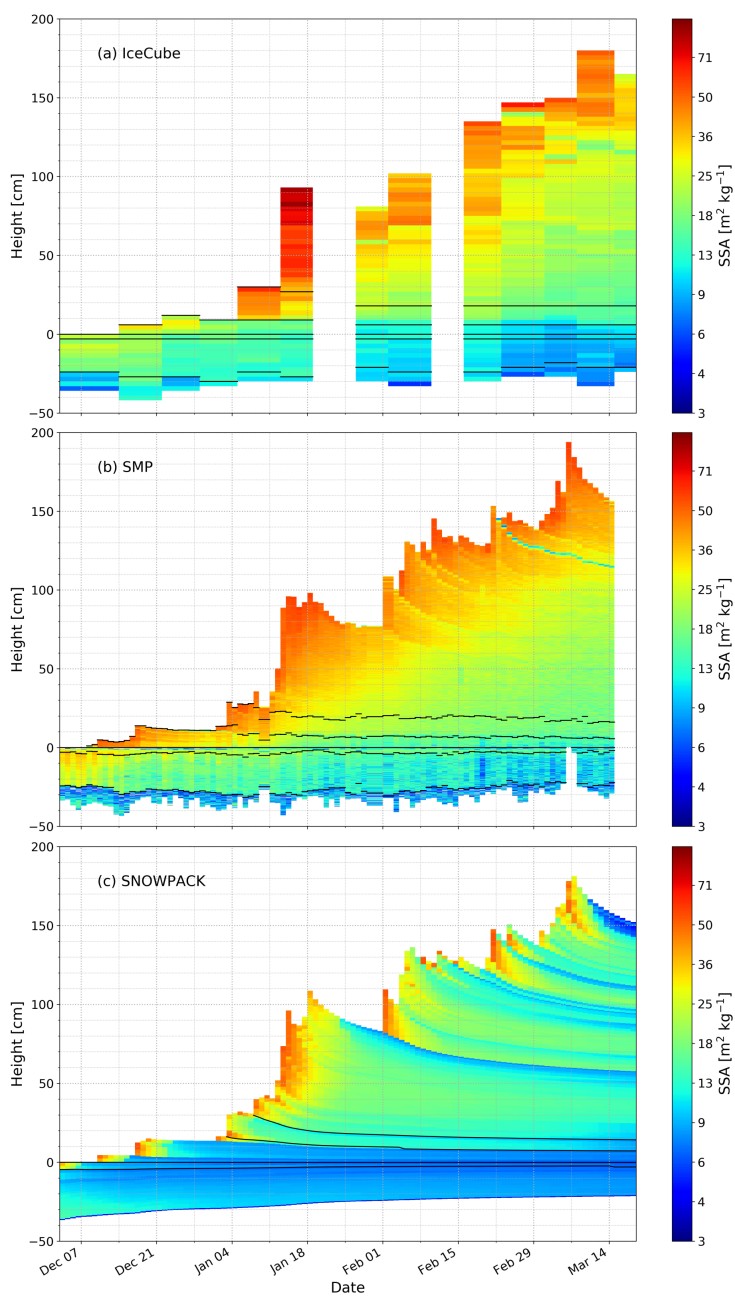

**Figure 9.** Evolution of the SSA profile during winter 2015-2016 (a) from IceCube measurements, (b) derived from SMP measurements, and (c) simulated by SNOWPACK. Boundaries shown with black lines allow identifying the 4 tracked layers as described for Figure 6.


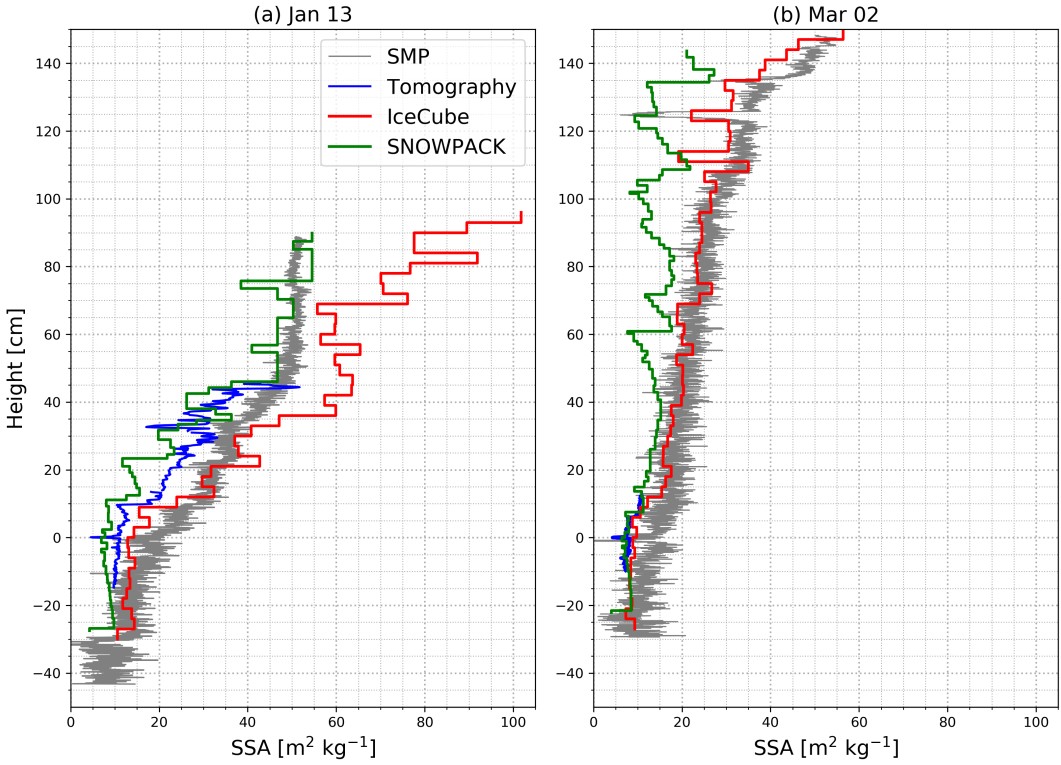

**Figure 10.** Vertical profile of SSA on 13 January 2016 and 02 March 2016 from SMP, IceCube, and tomography measurements as well as modeled by SNOWPACK.

averaged within boundaries of specific layers of the snowpack, as used in Wever et al. (2015) and in this study (Fig. 8 and 11). This latter method seems particularly suitable to assess the skill of parameterizations of internal snow processes, e.g. temporal evolution of density and SSA of a fresh snow layer or of a buried layer of surface hoar. Layer properties evolution are indeed very close to the formulation of equations in a Lagrangian model. The method ii) and iii) bear with uncertainties from vertical

5    mismatches that might contribute to the scatter between measurements and simulations and should thus be first corrected. When comparing paired-values at the same height, the prior alignment of the profiles is necessary. In the present case, we could simply re-align the profiles thanks to the presence of the dominant MF-layer in all measurement methods and throughout the season. Slight vertical mismatches can however be found. For example, the density profile of March 2, 2016 (Fig. 7) shows two distinct denser layers at around 125 cm and 135 cm height which are well identified in both SMP and density cutter measurements but

10   with a height mismatch of about 5 cm. This re-alignment method based on the identification of a persistent and well-defined snowpack feature might however not be always applicable. A more systematic approach could be the algorithm presented by Hagenmuller and Pilloix (2016) to automatically match snow profiles by adjusting their layer thicknesses. This methods has a strong potential for quantitative comparison studies (Hagenmuller et al., 2018). When comparing properties of specific layers, the definition of the layers boundaries is critical. The second-order fluctuations observed in the evolution of density and SSA

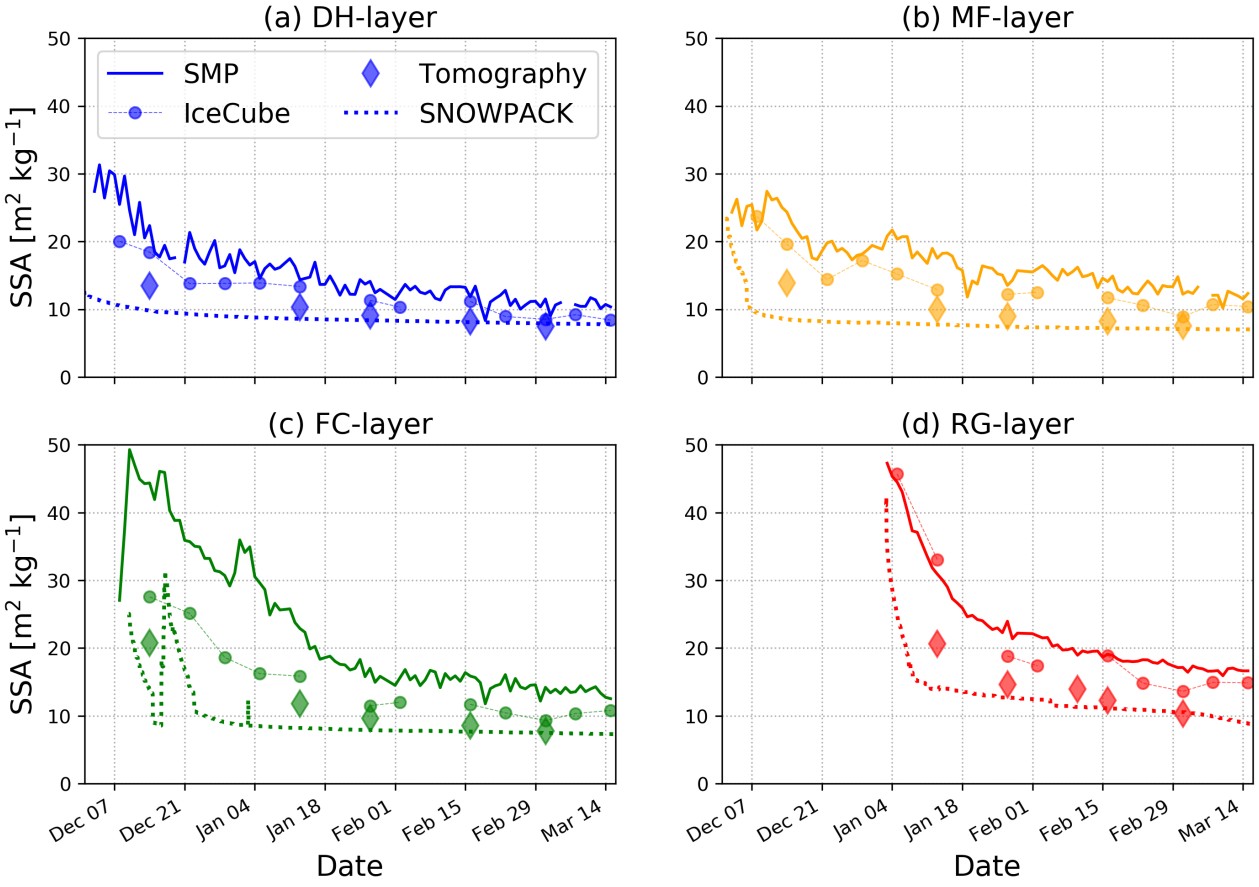

**Figure 11.** SSA evolution of the 4 tracked layers from SMP, IceCube, and tomography measurements as well as modeled by SNOWPACK.

of the MF-layer (Fig. 8 and 11), especially visible in the SMP data, might possibly result from the boundaries definition of this layer, in addition to the natural spatial variability of snow. Besides, the manual definition of boundaries is rather time-consuming if numerous layers are tracked. A more automatic method could be developed. In this respect, the RHOSSA data constitutes a valuable resource due to the continuity of the spatio-temporal picture of the seasonal evolution of stratigraphy.

## 7.2 The potential of daily SMP measurements

With daily SMP measurements, the RHOSSA campaign allows following the evolution of the internal structure of a snowpack at a sub-centimeter vertical resolution almost continuously over 4 months - up to now inaccessible. An unparalleled, smooth picture of the spatio-temporal evolution of density and SSA is revealed, contrasting with data from the classical snow pit measurements (Fig. 6 and 9). Many thin stratigraphic features are indeed clearly visible in the SMP data but only diffusely shown by the manual measurements. This highly detailed picture of the snowpack evolution opens new opportunities for field



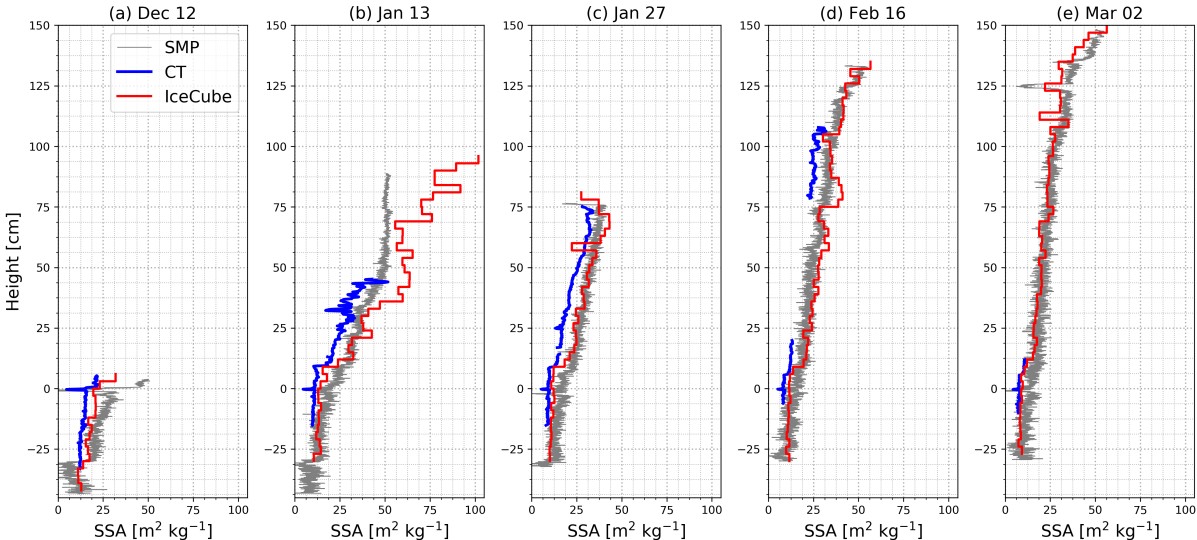

**Figure 12.** Comparisons of SSA profiles from tomography, IceCube, and SMP measurements.

studies on snowpack processes occurring over short-time scales (e.g. densification of fresh snow) or very localized (e.g. rain crust or surface hoar formation), as well as refined evaluation of snow models as already mentioned.

   One advantage of SMP measurements compared to snow pit measurements is they are relatively faster (of the order of 30 minutes for five measurements) and thus more suitable for daily snowpack monitoring. It is however important to keep in mind

that density and SSA are not directly *measured* by the SMP but *derived* from the force signal based on parameterizations (Fig. 2), bearing additional uncertainties comparing to other more direct measurements. Several parametrizations were previously put forward to derive density and/or SSA from SMP signals (e.g. Pielmeier and Schneebeli, 2003; Dadic et al., 2008; Proksch et al., 2015; Kaur and Satyawali, 2017). Differences between the parameterizations of density and SSA of Proksch et al. (2015) and the ones presented in this study are likely due to the version of the SMP device which has undergone an update of the

electronics in version 4 that affected the inversion of the model Löwe and van Herwijnen (2012) through the force correlation function. We would hope that the parameterization Eq. (1) and (2) are generally applicable to an SMP version 4. However, without an independent validation by measurements under different snowpack conditions, it is not possible to state the range of validity of the parametrizations presented here. In the long term, it would be desirable to improve the underlying stochastic-mechanical approach (Löwe and van Herwijnen, 2012) by an invertible model that contains density and SSA to retrieve these

parameters from a more physical picture of the penetration process.



### 7.3 Comparing density and SSA estimates

As possible starting points to future dedicated studies, we sum up here the main deviations reported in this paper when comparing density and SSA estimates. First, we recall that density and SSA derived from SMP data were obtained to best match results from the cutter and IceCube measurements, so they necessarily inherit their performances.

We report a significant and systematic inter-measurement deviation in the SSA estimates. Values from IceCube and SMP are systematically higher than values computed on tomographic images, approximately by a factor 1.3 (Fig. 12). A comprehensive comparison between optical methods, such as IceCube, and tomography seems very much needed to understand this systematic deviation. Besides, large disagreements were reported on the specific day of January 13, 2016, for which IceCube data range from 60 to 100 $m^2$ $kg^{-1}$ whereas SMP data show values around 50 $m^2$ $kg^{-1}$ (Fig. 12b). That day, measurements were

performed during a snowfall in light freshly-deposited snow. When measuring SSA of light snow, typically for values above 60 $m^2$ $kg^{-1}$, the emitted radiations can interact with the bottom of the sample holder during the measurement causing an overestimation of the SSA (Gallet et al., 2009; Zuanon, 2013). Another possible cause is that the present statistical model used to derive SSA from SMP measurements fails to reproduce the high SSA values of newly-deposited snow because of their under-representations (one day) in the IceCube dataset used for calibration; (similarly but to a lower extent, disagreements are

found in the upper 20 cm of the density profiles of the same day (Fig. 7a): SMP measurements fail to capture the very low density measured by the cutter method (60 $kg$ $m^{-3}$ vs. 30 $kg$ $m^{-3}$)).

Comparing SNOWPACK outputs against observations, one significant deviation is the absence of the MF-layer in the simulations. This is due to the fact that the precipitation forcing scheme used in the present simulations does not allow representing rain fall events occurring at negative air temperatures. This inappropriate forcing could be improved by using diagnostic at-

mospheric variables to detect such events Quéno et al. (2018). Also, SNOWPACK underestimates SSA in overall (Fig. 9, 10, and 11). A similar bias was reported at an arctic site (Leppänen et al., 2015). On the contrary, a systematic overestimation of the SSA simulated by Crocus was recently pointed out (Tuzet et al., 2017). Evaluations can however be challenged by the significant inter-measurement deviations observed, as discussed above. The agreement between simulations and estimates from tomography is better than between simulations and estimates from SMP or IceCube. Finally, SNOWPACK seems to slightly

overestimate the densification rate of persistent weak layers, as observed in our study for the DH-layer and FC-layer (Fig. 8a and Fig. 8c). Barrere et al. (2017) reported similar findings with the model Crocus. The discrepancies pointed out here suggest further investigations and might guide possible model improvements.

### 8   Conclusions

During winter 2015-2016, the standard snow observation program of the WFJ site (Eastern Swiss Alps, elevation 2536 m) was

complemented by additional measurements and stability tests, bridging between traditional and newly-developed measurement methods. This campaign results in a multi-resolution and multi-instrument dataset of structural and mechanical properties of the snowpack, referred as the RHOSSA dataset. The dataset includes time-series of density, SSA, and critical cut length, traditional snow pit parameters, and results from compression tests. Profiles of density and SSA were monitored daily and with



a vertical resolution of 1 mm based on SMP measurements. These high-resolution data offers an unprecedented smooth and continuous picture of the snowpack evolution throughout the season.

Our specific results comprise (i) re-calibrated parameterizations to estimate density and SSA from SMP measurements for version 4, (ii) the comparison of density and SSA estimates from state-of-the-art measurement methods (Cutter/IceCube, tomography, SMP-derived), and (iii) the assessment of the SNOWPACK model against measurements. Results from the two latter point contribute describing current states and suggesting further investigations. Our study demonstrates the potential of high temporal and spatial resolution dataset for the evaluation of the detailed snowpack models as Crocus or SNOWPACK. In this view, the RHOSSA measurements campaign could be extended to other snow observation sites to cover different environments and conditions.

## 9 Code and data availability

The dataset presented in the paper will be available on the EnviDat database (doi will be provided).

*Author contributions.* All authors contributed to the field measurements. B.R., H.L., and N.C. wrote the paper with input from C.F. and A.V.H. A.V.H., B.R., C.F., H.L., J.S., and N.C. performed the analysis of the data and the simulations. N.C. and M.S. directed the project.

*Competing interests.* The authors declare that they have no conflict of interest.

*Acknowledgements.* We greatly thank Lino Schmid (SLF), who performed some of the traditional profiles, and Margret Matzl (SLF), who did the $\mu$CT scans presented in this study. We would like to acknowledge Martin Proksch and Ben Reuter (SLF), who have initiated the project of daily SMP measurements. We also greatly thank the PhD students of the SLF for their contributions in various snow measurements during winter 2015-2016. N. Calonne was supported by grant 152845 from the Swiss National Science Foundation and by the WSL-project 201612N1411.



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
