# Peer review of "The RHOSSA campaign: Multi-resolution monitoring of the seasonal evolution of the structure and mechanical stability of an alpine snowpack"

_The Cryosphere, 2019_

## Referee Comment (RC1) · Joshua King (Referee) · 9 Jan 2020

The authors present a local-scale study aimed at characterizing seasonal snowpack evolution with traditional sampling (snow pits), advanced techniques (SnowMicroPen, IceCube, and Tomography) and model application (SNOWPACK). Applying a multi-scale approach, methods are intermixed to construct a daily time series of vertical variation in snow density and specific surface area. The methods are cross-compared to contribute a recalibration of the Proksch et al. (2015) SMP empirical model and to evaluate SNOWPACK simulations. Analysis of the dataset demonstrates clearly how recent advances in field methodology can support model evaluation at very high vertical resolutions. In particular, the details found in Figures 6 and 9, where SMP derived snow properties are introduce at daily time steps, show ability to track snow events and metamorphosis captured in SNOWPACK simulations. Overall, the paper provides a great summary of the campaign results and demonstrates how future model evaluations can befit from applying similar seasonal framework.

Prior to publication, the paper would benefit from some restructuring to clarify properties of generated the dataset and promote repeatability. These would be meaningful additions to allow application of this work to other environments:

- Recalibration of the Proksch et al. (2015) model uses collocated SMP profiles and density cutter measurements. No distinction is made between the training and testing data when evaluating Eqns 1 or 2. If the authors felt cross-validation was unnecessary, please include this information so that the reader can determine if the skill estimates may be biased (i.e. Test-Train are identical datasets).

- I'd like to better understand why realignment resulted in improved correlation between the cutter/IceCube measurements and SMP derived properties in Figure 2 as indicated in text (P9 L23). If alignment with the persistent layer defined in Section 6 resulted in a better vertical matching, why were the better alignments not used for the initial recalibration? Throughout the paper, descriptions of alignment could be improved and are noted in the extended comments below.

- While the layer tracking analysis is meaningful (Fig 8 and 11), description of the SMP tracking method is difficult (if not impossible) to reproduce. An enhanced description of how transitions in SMP signal were used to define layers would be a helpful addition.

- I can confirm that the revised coefficients presented for SMP density are improved over those Proksch et al. 2015 for Arctic snow and snow on sea ice. However, local calibration with our SMP4 unit resulted in quite different coefficients and better RMSE over the use of global parameters (P23 L11). This may make it important to make clear thecalibration methods so that they can be easily repeated for different environments

or units(?).

If length of the paper becomes an issue, the authors may consider revising or removing stability content (i.e Figure 5). Stability metrics are not used in the context of model comparison, which carries as a central theme. However, I've not made any substantive comments on stability analysis itself as this is beyond my expertise. Often lost on readers of field-based papers is the incredible amount of work that goes into collecting high quality datasets. Congratulations to the authors on executing this ambitious work, and I sincerely hope that the community can benefit from the dataset and methods presented in this paper.

Josh

The following are general comments and suggestions for consideration: P2 L5 – Suggest removing the 'e.g' and revising as 'data back to 1936 in the case of WFJ'.

P2 L8 – Please be explicit about which properties are characterized rather than using 'hard hardness . . ..'.

P2 L9 – Remove the period between the citation and sentence.

P2 L14 – Can you clarify what 'non-empirical snow properties' means? This statement is unclear.

P2 L15 –Ideally traditional measurements would be supported with metrics such as SSA but the use of the word 'tends' seems to imply this IS a frequent practice. Could it rephrased with the word 'can' or similar?

P2 L19 – Capitalize 'IRIS'. Stands for 'InfraRed Integrating Sphere'.

P3 L16 – Should the word 'such' be in this sentence?

P3 L21 - It feels a bit discouraging to say that the stated goals are dependent on availability of a large dataset with many tools. As a suggestion, removing the word 'only' might lessen the tone. The wording 'cross-validation' could also be problematic

as it refers to a specific statistics method. Later the wording 'cross-comparison' (P4 L8) is used which seems to be a better fit.

P4 L12 – Degree symbols should accompany the coordinate units.

P4 L14 – Consider revising the sentence to mention dry snow conditions only once.

P4 L23 – The second element of the measurement area description is squared. Was this intended?

P7L7 – If the Zuanon (2013) methods were adopted, were any samples compressed to avoid over penetration of the laser? A sentence on how samples were extracted and prepared would be useful for future comparisons where this has become common practice.

P7 L8 – What about uncertainty with low SSA (i.e. DH or FC)? Standard deviation of the measurements in Figure 10a appears to increase with depth and is quite large relative to tomography.

P7 L17 – Would like to see an enhanced description of what goes into the profile quality check. Previous studies have described linear trends while measuring in air while others have provided quantitative methods to apply a noise threshold. Which approach was used to determine drift or accept/reject a profile?

P7 L20 – What were the qualities of the data, snow, or study site that determined the profiles could be matched without an offset correction? In section 6 the opposite seems to be stated that spatial variability required compensation to avoid height mismatches (P11 L13).

P7 L29 – Suggest removing 'Reconstruction followed standard procedure' as it's described in the next sentence.

P8 L10 – May be helpful to indicate the rate of replacement.

P9 L7 to 11 – Found this a bit of confusing. Is the single 'median' profile being used to

train (1)? Perhaps the alignment sentence could be moved upwards in the paragraph to clarify. As it reads now I was not able to determine if 1 profile per pit is being used or if multiple A-S aligned and cropped profiles are being used.

P9 L15 – Please provide the number of compared measurements to support of the significance test.

P9 L16 – This differs substantially from Proksch et al (2015) where coefficients for SSA were not provided. This new equation requires no estimate of density from the SMP, which arguably is better if SSA is the target (minimizes bias from density coefficients and conversion from d0?). No action to take unless the authors wish to highlight the benefit of avoiding the conversion of L_ex to SSA.

P9 L23 – An enhanced explanation of why the values in Figure 2 do not reflect the error/skill assessment in this section is needed. Related questions: Why does correlation improve when Eqn. (1) was trained on a different set of comparisons? Why was Eqn (1) was not just trained on this better alignment to begin with?

P8L29 – Remove one set of brackets around the Eqn.

P10 L2 – What was the statistical test that showed the boundary transition to be significant? If untested, consider removing the word 'significant'. See comments in the initiate statement about repeatability as well.

P10 L3 – Given that the boundaries were identified subjectively, will their heights be provided in the published dataset?

P15 L3 – I agree that the information is really useful to show the formation and evolution of these fine features. However, given that Figure 6b has no minor or major ticks for the initial date (Feb 22) it's fairly difficult to identity the feature. Could a label be provided for easy reference?

P23 L11 – I can confirm that the recalibrated density coefficients don't produce a best-possible estimates of snow density with our SMP for Arctic snow. Would be very interesting to combine datasets from multiple units to evaluate this uncertainty.

P25L20 –Citation style should be a paraphrase.

Table 1 - List the number of measurements as a separate column. The large number of measurements is really smoothing to highlight! This will also be helpful in the future to frame comparison.

Figure 2 - Add N, Rˆ2 and RMSE be added to these diagrams. Having a quantitative evaluation in the diagram provides a quick reference for the reader.

Figure 4 – Please provide a colour legend for the grain type classifications even though they are standardized. Additionally, is it possible to provide sub-hatching for the hand hardness levels? It's challenging to determine the level past the first data.

Figure 6/9 – Has the SMP data been smoothed or aggregated? This does not appear to be mentioned in text but Figure 10 shows variability in SSA absent in Figure 9 at the 1 mm scale.

---

## Referee Comment (RC2) · Anonymous Referee #2 · 12 Jan 2020

The paper presents the RHOSSA campaign focusing on snow density, SSA and stability measurements over one winter in Weissfluhjoch, Switzerland. Modern methods such as SMP and IceCube are compared with traditional snow pit measurements and SNOWPACK modeling. Measurement results demonstrate how modern methods can increase temporal and vertical resolution in snow profiling compared with traditional measurements. This kind of data sets allow proper evaluation of modeling results, which is not possible using traditional measurements due to their poor temporal and vertical resolution. The main result is the recalibration of Proksch et al. 2015 model for deriving SSA and snow density from SMP data.

[Figure]

The snow stability part is a bit disconnected from the main text, which focuses on SSA and density. The authors could consider dropping the stability measurements.

Specific comments:

p4r4 Section 6-> Section 5

p4r12 Degrees missing from coordinates.

p6r15 The snowpack was sampled with 3 cm resolution. What did you do with layers thinner than 3 cm? This explains why the 22 Feb layer is "only diffusely reported in the IceCube data" (p16 r17), if it is mixed with grains from other layers. How did you sample the MF layers? They are very difficult to get into sample holder without breaking them. Were the low density layers compacted to avoid measuring the sample holder?

p8r8 Why exactly 1.2 °C?

p9r10 What is the justification for selecting different method for matching the profiles here than later in the paper (p9r24)? If re-aligning profiles using the MF layer resulted in "better correlation between estimates from SMP and snow pit measurements", why didn't you use the same method here to derive the parameters?

p9 The model parameters are derived from IceCube measurements. Later (e.g. Fig 11) you show that there are big differences between IceCube and tomography measurements. Please comment on the accuracy of SMP-derived SSA values.

p11r2 choose->chose

p11r20 caption->panel

p22 Fig 11. The difference between SSA derived from SMP and tomography varies between different layers. Do you think the snow structure (grain type) has something to do with that? Should the SSA model be calibrated separately for different grain types? And why are there big differences between IceCube measurements and SMP, if IceCube data was used in the fitting, shouldn't they agree better?

p23 Fig 12. Please add SNOWPACK profiles as well.

---

## Referee Comment (RC3) · Alexandre Langlois (Referee) · 30 Jan 2020

General comments: The paper highlights results from a winter field campaign based out of the well-known WFJ site in Davos. The authors present a temporal analysis of snow microstructure and mechanical properties using state-of-the-art instruments that all have their advantages and limitations. Of particular relevance, repeated SSA and resistance measurements using an IceCube and the SMP are presented and compared against SNOWPACK simulations. The originality of the paper reside in a new calibration for the V4 of the SMP that will be indeed useful for international users such as my own group.

Overall, the paper is clearly written, with a very thorough analysis that certainly is worthy of publications. The expertise and reputation of the author's list is obviously excellent. I however, have several comments and questions that I would like to see addressed from my own perspective of being a SNOWPACK user in the Arctic with our own SMP and IRIS instrument since I think some elements need stronger analysis or at least physical explanations from the results presented in the paper given that very important science questions remain open.

Specific comments: In general terms, using SNOWPACK is not trivial. Yes the model can run virtually anywhere, especially in Switzerland where it was developed but certainly harder elsewhere. A realization we came with as being users since 2002 is that the model remains very sensitive to 1) forcing dataset, 2) soil configuration and 3) obviously the internal physic calculations of microstructural elements that have changed from version to version over the years. For instance, a bias is observed in Canada on snow depth as a function of precipitation rate; or again bias in microstructure are not the same given the metamorphic process in place (kinetic vs equilibrium). Section 4 of the paper present the model in very general terms, I would suggest modifying this section to: SNOWPACK configuration where the authors would list: better description of the meteorological forcing dataset; soil configuration (type, roughness, how many soil layers?). There is also no mention of the spin up? Was the simulation initiated with a snow profile? It is obvious form the author list that the simulation is more than likely to be well parameterized, simply that I think there are more and more SNOWPACK users aware of potential problems, so more details on the simulation configuration I think would be very beneficial.

Page 2, Line 8: 'spatially consecutive'. . .what is meant exactly? A clarification be appreciated. I assume the snowpit in such a confined space is useful for time-series, to avoid any variability due to spatial variability processes.

Page 2, Line 18: I would argue to add as a more general term the importance in surface energy balance, which in turn plays a critical role in freeze-thaw cycles for example. So

the importance for large scale processes.

Page 3, Line 5: . . .change to gap in temporal resolution

Page 3, Line 17-18: This was the whole idea behind the Snow Grain workshop held several years ago. Would the authors consider revisit some of the data?

Page 3, Line 29: how were selected the sites? It is mentioned that site were chosen on 'selected locations' but we all know site selection is critical. Some details on how the sites/samples were chosen be appreciated.

Table 1: add units to the measured/derived properties.

Page 5, Line8: ECT are extended column test, not extended compression test.

Section 3.2.: What was used to weigh the density cutter?

Page 7, Line 8: The 10% ucertainty is for IceCube or DUFISSS? IceCube was used, but the reference provided is for DUFISSS. What is the published accuracy of IceCube?

Page 8, Line 8: I know the 1.2C threshold is used, likely well parameterized for WFJ. However I assume mixed precipitations are possible, what uncertainty can arise from such cases? A study by Ding et al. (2014) suggest that precipitation type are not only a factor of Tair, but also altitude and relative humidity. So how precise, at WFJ is precipitation phase parameterized?

Section 5.1: Our group is also doing just that with our own SMP this winter. Our concern is, that we are working on deriving a SMP(lc) method based on vertical 'z-axis' measurements from the SMP, with IceCube and density cutter that have a strong 'y-axis' component. We are asking ourselves if the SMP 'F' and 'L' parameter would be the same if we were to conduct a SMP profile in the 'y-axis' (i.e. in H instead of V). . . From an anisotropy point of view, I think we can expect them to be different. Also we have an IceCube that includes a very thin layer being samples, with a density cutter that include a lot more snow. . . We are dealing with different scale, yet trying to correlate

them together, I am fully aware that for now, this is the way to do it. Simply that I'd be happy to hear the authors ideas on this offline.

Section 6.1.: the problems linked to the vapor flux parameterization behind the growth of depth hoar is well known (Domine et al, 2019; Gouttevin et al., 2018). I'm also aware of the current work done in author's lab to correct that problem. Given the temperature and snow depth stated, yes I'm not surprise to see presence of depth hoar. Although, I'm pleased to see that SNOWPACK seems to react quite well to this, especially when I'm looking at Figure-7 where the depth hoar layer is indeed corresponding to a reduction in density as can be expected. This was a problem, that now looks much better. So my question is: did the authors used a different metamorphism parameterization to reach this? Or the standard version online was used without further modification?

Figure-4 would be much easier to read with a legend.

Page 16, Line 2: Why does SNOWPACK overestimate the density of the DH layer? Is it because of the absence of vapor flux from the ground leading to the underestimation of the SSA?

Section 6.3: When using IceCube, it is very hard to sample properly depth hoar by the simple nature of the thickness of the hoar layer vs the sampler size. Any sampling difficulties were encountered using IceCube in these conditions?

Section 7.1, Lines 9-10: I would argue that yes there is a range, but it remains alpine where the processes governing stratigraphy, energy transfer is a different world from what we find in the Arctic, or even in other alpine regions of the world. I would argue to state that the snowpack offered a wide range of alpine snow conditions.

Section 7.2.: With a snowpack having a temperature gradient important enough to lead to the formation of a depth hoar layer, can expect to have a decent variability in temperature vertically obviously. But, the effect of changing temperature as the SMP travels through snow is not discussed. I know the authors are aware of this problem,

can they confirm this was not an issue in this environment?

Page 24, Line 12: I think it is more a problem of the laser hitting the side of the sampler rather than the bottom, but this is a small detail.

Again, this is a very nice contribution made by a very solid team at a site internationally known. I would suggest my comments to be minor, and would be happy to see this work published after the comments above are addressed.

---

## Author Comment (AC1) · 10 Apr 2020

**Response to Review 1**

We thank very much Reviewer 1 for his comments that help improving the manuscript. Please find below our point-by-point replies in blue color.

The authors present a local-scale study aimed at characterizing seasonal snowpack evolution with traditional sampling (snow pits), advanced techniques (SnowMicroPen, IceCube, and Tomography) and model application (SNOWPACK). Applying a multi- scale approach, methods are intermixed to construct a daily time series of vertical variation in snow density and specific surface area. The methods are cross-compared to contribute a recalibration of the Proksch et al. (2015) SMP empirical model and to evaluate SNOWPACK simulations. Analysis of the dataset demonstrates clearly how recent advances in field methodology can support model evaluation at very high vertical resolutions. In particular, the details found in Figures 6 and 9, where SMP derived snow properties are introduce at daily time steps, show ability to track snow events and metamorphosis captured in SNOWPACK simulations. Overall, the paper provides a great summary of the campaign results and demonstrates how future model evaluations can benefit from applying similar seasonal framework.

Prior to publication, the paper would benefit from some restructuring to clarify proper- ties of generated the dataset and promote repeatability. These would be meaningful additions to allow application of this work to other environments:

- Recalibration of the Proksch et al. (2015) model uses collocated SMP profiles and density cutter measurements. No distinction is made between the training and testing data when evaluating Eqns 1 or 2. If the authors felt cross-validation was unnecessary, please include this information so that the reader can determine if the skill estimates may be biased (i.e. Test-Train are identical datasets).

→ The entire cutter and IceCube data have been used to "train" the SMP data and to obtain Eq. (1) and (2). The scatter plots shown in Figure 2 show the quality of these parameterizations for the same dataset, i.e. SMP derived data from Eq(1) and (2) versus cutter and IceCube data. The "Train" and "Test" dataset are thus the same. We aimed here at getting as close as possible to this particular cutter and IceCube dataset from our SMP data, and we did not evaluate the obtained parameterizations with other independent dataset. This is why we wrote page 23 line 10: "We would hope that the parameterization Eq. (1) and (2) are generally applicable to an SMP version 4. However, without an independent validation by measurements under different snowpack conditions, it is not possible to state the range of validity of the parametrizations presented here."

We improved Section 5.1 so that it appears more clearly that the test and train data are the same, p10, L6: "This plot [Figue2] shows the observed density from cutter measurements against the SMP-derived density obtained from Eq. (1) and from Proksch2015 for the 15 days for which both data are available (same dataset as used for the statistical modeling). Similarly, the observed SSA from IceCube measurements are presented against the SMP-derived SSA from Eq. (2) and from Proksch2015 for the 13 days for which both data were available (same dataset as used for the statistical modeling). To do so, and as done for the statistical modeling, SMP-derived properties were averaged over 3 cm resolution and SMP and snow pit profiles of the same day were re-aligned with the snow surface and cropped to the length of the shortest profile."

- I'd like to better understand why realignment resulted in improved correlation between the cutter/IceCube measurements and SMP derived properties in Figure 2 as indicated in text (P9 L23). If alignment with the persistent layer defined in Section 6 resulted in a better vertical matching, why were the better alignments not used for the initial recalibration? Throughout the paper, descriptions of alignment could be improved and are noted in the extended comments below.

→ We thank the reviewer for pointing out this issue in the paper. We agree on the confusion about the alignment. For explanation, we would like to point out the difference between 1/ matching of profiles *of the same day* for statistical analysis, and 2/ matching for visualisation of the data such as the evolution of *profile with time*.

1/ Alignment of co-located, co-temporal profiles can be done by using the snow surface. This is convenient and always applicable (unlike using a specific layer) so it is a suitable method to use when doing a local re-calibration of the SMP parameterizations as in our study.

2/ Alignment of profiles when plotting their evolution with time requires another method of matching since profiles are then not co-temporal and do not share a common height/snow surface. One way is to re-align profiles with the ground. For sites showing a ground that is uneven or bumpy this method can however lead to a mediocre alignment. This was the case of the WFJ (ground is uneven) and we found out that a re-alignment based the crust MF-layer offers a qualitatively better match, when looking at plots of Fig 6 and 9 for example. Hence, we chose this alignment method for to present data in Figure 6, 7, 9 and 10.

→ As pointed out by the Reviewer, the first version of the paper showed an inconsistency related to the choice of the alignment method in Section 5.1. Indeed, method 1 (snow surface alignment) was used to develop the statistical model but method 2 (MF-layer alignment) was used to test the performance of the model (Figure 2).

We fully agree with the reviewer that it is confusing. Thus, we modified so that method 1 (snow surface alignment) is now used for both the statistical model and the analysis of the model performance. Method 2 (layer alignment) is only used later in the paper, in the Result part, for time-series plotting purposes.

Modifications throughout the paper have been done accordingly, especially:
-   Figure 2 has been redone, based on data re-aligned using the snow surface
-   $R^2$ coefficients associated to Figure 2 have been modified. They are slightly better than the previous version (from layer alignment to snow surface alignment: $R^2$ changes from 0.73 to 0.75 for density and from 0.81 to 0.82 for SSA, using Eq 1 and Eq 2 respectively). This actually makes sense as Eq. 1 and 2 have been developed from data aligned with the snow surface.
-   Section 5.1 reads now, p10, L6: The performance of the new parametrizations compared to the original parametrizations of Proksch2015 is presented in Figure 2. This plot shows the observed density from cutter measurements against the SMP-derived density obtained from Eq. (1) and from Proksch2015 for the 15 days for which both data are available. Similarly, the observed SSA from IceCube measurements are presented against the SMP-derived SSA from Eq. (2) and from Proksch2015 for the 13 days for which both data were available. To do so, and as done for the statistical modeling, SMP-derived properties were averaged over 3 cm resolution and SMP and snow pit profiles of the same day were re-aligned with the snow surface and cropped to the length of the shortest profile. "

- In the introduction to the Result part, p12, L3, we included now: "To present the evolution of profile properties with time, vertical profiles presented in the following were re-aligned such as z = 0 cm corresponds to the height of the upper boundary of the MF-layer (i.e. the 20151202-boundary). Choosing this layer as a height reference leads to a qualitatively better match than by simply taking the ground as reference (the field site ground at WFJ is uneven)."

- While the layer tracking analysis is meaningful (Fig 8 and 11), description of the SMP tracking method is difficult (if not impossible) to reproduce. An enhanced description of how transitions in SMP signal were used to define layers would be a helpful addition.
→ Section 5.2 "Layer tracking" has been restructured and some reformulation has been made to improve the description of the method. Layers in SMP data were tracked in the same way as in the cutter and IceCube data, i.e. by a manual identification of boundaries in the snow property profiles. The paragraph now reads: "In the measurements data, the layers of interest were defined by the height of their upper and lower boundaries. Boundaries were manually identified by simply looking at the property profiles, looking for sharp and relevant transitions, and recording heights. This step was performed on all the weekly density profile from the cutter and SSA profile from IceCube, as well as on all the daily representative profile of penetration force resistance obtained from the five daily SMP measurements. The identification of layer boundaries was sometimes challenging for weak stratigraphic transitions, e.g. the transition between a layer of fresh snow that fell onto a soft snow layer. To help in such cases, boundaries could be backtracked in time, starting from a profile where the layer of interest is older and its boundaries more clearly detectable. Also, additional information, such as observed height of new snow, was sometimes used to help delineate boundaries."
Besides, we would like to point out that this method only works when tracking well-pronounced layers, so might be hard to use in a systematic way over entire snowpack profiles. To stress this point, we added p10, L23: "The first step is to define which are the layers of interest, knowing that this method is only possible with layers that contrast well enough with their surrounding, so their boundaries can be identified by a significant and rather sharp transition in the vertical profile of snow properties."

- I can confirm that the revised coefficients presented for SMP density are improved over those Proksch et al. 2015 for Arctic snow and snow on sea ice. However, local calibration with our SMP4 unit resulted in quite different coefficients and better RMSE over the use of global parameters (P23 L11). This may make it important to make clear the calibration methods so that they can be easily repeated for different environments or units(?).
We improved the description of the calibration method in Section 5.1, making sure that each step is clearly described.

General comments

P2 L5 – Suggest removing the 'e.g' and revising as 'data back to 1936 in the case of WFJ'.
→ Modified accordingly

P2 L8 – Please be explicit about which properties are characterized rather than using 'hard hardness . . ..'.

→ We modified accordingly; it reads now "grain size, grain shape, hand hardness, and wetness" (P2, L8).

P2 L9 – Remove the period between the citation and sentence.
→ Modified

P2 L14 – Can you clarify what 'non-empirical snow properties' means? This statement is unclear.
With "non-empirical properties" we refer to properties that are physically/mathematically-defined, such as density and SSA, in contrast to grain shape for instance which has no mathematical definition. We modified the term and use "objectively-defined snow properties" (P2, L15).

P2 L15 –Ideally traditional measurements would be supported with metrics such as SSA but the use of the word 'tends' seems to imply this IS a frequent practice. Could it rephrased with the word 'can' or similar?
→ Modified accordingly. The sentence reads now "Concerning the characterization of snow microstructure, the observer-biased estimate of traditional grain size can be replaced by measurements of specific surface area" (P2, L15).

P2 L19 – Capitalize 'IRIS'. Stands for 'InfraRed Integrating Sphere'.
→ Modified accordingly.

P3 L16 – Should the word 'such' be in this sentence?
→ We modified the sentence as "These examples exploit key advantages of the SMP, namely fast profiling for frequent measurements and high vertical resolution, so that profiles are obtained at a considerably finer scale (mm) than possible with traditional means." (P3, L17).

P3 L21 - It feels a bit discouraging to say that the stated goals are dependent on availability of a large dataset with many tools. As a suggestion, removing the word 'only' might lessen the tone. The wording 'cross-validation' could also be problematic as it refers to a specific statistics method. Later the wording 'cross-comparison' (P4 L8) is used which seems to be a better fit.
→ We agree with the reviewer and modified the sentence accordingly as "In the context raised above, the value of emergent, objective snow properties, their potential to replace traditional means in operational snow monitoring programs, and their requirements on temporal and vertical resolutions for model evaluations can be investigated within a multi-resolution and multi-instrument dataset to facilitate comprehensive cross-comparison analyses."

P4 L12 – Degree symbols should accompany the coordinate units.
→ Modified accordingly

P4 L14 – Consider revising the sentence to mention dry snow conditions only once.
→ Modified as follows "We focused on the period from beginning of December 2015 to end of March 2016 to ensure measurements in dry snow condition as required by some of the used instruments. (P4, L17)

P4 L23 – The second element of the measurement area description is squared. Was this intended?
→ Modified as follows "20 m x 8 m".

P7L7 – If the Zuanon (2013) methods were adopted, were any samples compressed to avoid over penetration of the laser? A sentence on how samples were extracted and prepared would be useful for future comparisons where this has become common practice.
→ The extraction of the sample was performed following the protocol described in Zuanon et al. 2013. In addition, we indeed systematically slightly compressed the extracted sample.  We included this information in the paper: p7, L10: "Snow samples were very slightly compressed when inserted into the sample holder and attention was paid to have a flat snow sample surface."

P7 L8 – What about uncertainty with low SSA (i.e. DH or FC)? Standard deviation of the measurements in Figure 10a appears to increase with depth and is quite large relative to tomography.
→ As pointed out in the paper Section 7.3, we report a significant and systematic inter-measurement deviation in the SSA estimates. Although we did not study in details uncertainty of SSA measurements in weak layers, our results do not show that biases are more pronounced for DH or FC layers. We did not observe an evolution of the bias with depth. The paper however stresses that these inter-measurement deviations should be further investigated.

P7 L17 – Would like to see an enhanced description of what goes into the profile quality check. Previous studies have described linear trends while measuring in air while others have provided quantitative methods to apply a noise threshold. Which approach was used to determine drift or accept/reject a profile?
→ We improved the description of the SMP data processing. The paragraph now reads P7, L21: "The quality control of SMP force profiles was done manually by rejecting signals with 1) visible trends either in the air portion of the signal or over the entire depth, 2) high noise levels and unrealistic spikes, and 3) frozen tip problems revealed by a force response that appears to be activated only deeper in the snowpack. Most of these problems are caused by wet conditions.  The air-snow and snow-ground interface were detected manually to remove air and ground regions from the signal."

P7 L20 – What were the qualities of the data, snow, or study site that determined the profiles could be matched without an offset correction? In section 6 the opposite seems to be stated that spatial variability required compensation to avoid height mismatches (P11 L13).
→ This seems to be a misunderstanding. We improved the description of the SMP data processing in Section 3.4. By offset correction we mean that the value of the force signal itself was not shifted by a given value as it can be sometimes observed (see previous comment). The force signal in the air was very close to zero (manual check) so we did not correct the force signal. This has no link with the height alignment performed in Section 6 for data visualisation.

P7 L29 – Suggest removing 'Reconstruction followed standard procedure' as it's described in the next sentence.

→ Modified accordingly

P8 L10 – May be helpful to indicate the rate of replacement.
→ During the period shown in this study (no melt out), only missing values of either incoming or outgoing SW or albedo values above 0.95 require a replacement. There are no missing values and the latter amount to at most 0.8%, predominantly at sunrise and sunset.

P9 L7 to 11 – Found this a bit of confusing. Is the single 'median' profile being used to train (1)? Perhaps the alignment sentence could be moved upwards in the paragraph to clarify. As it reads now I was not able to determine if 1 profile per pit is being used or if multiple A-S aligned and cropped profiles are being used.
→ We agree with the reviewer and modified the paragraph to describe more clearly each step of the process. It reads now, P9, L16: "The statistical modeling was applied based on a sub-dataset of data from the days for which both SMP and snow pit measurements were available (15 days for density, 13 days for SSA). From each raw force signals, parameters F and L were computed from the raw penetration force profiles over a sliding window of 1 mm with 50% overlap, yielding profiles of F and L with a vertical resolution of 0.5 mm. Note that Proksch et al. (2015) used a sliding window of 2.5 mm, but tests with different window heights (1, 2.5 and 5 mm) did not show a significant impact. Next, for each day, the five daily profiles of F and L of the same day were aligned by simply using snow surface as common reference and a median operation was applied to get one representative profile of F and L per day, called the median profiles in the following. Next, each median profile was averaged vertically using a 3 cm window to match the vertical resolution of the snow pit measurements. Finally, the median 3cm-averaged profiles F and L and the profiles of rho_cutter and SSA_ic of the same day were aligned by using snow surface again as common reference and cropped to the length of the shortest profile. This way, all profiles of a given day are described on the same vertical scale and values of F, L, rho_cutter and SSA_ic can be paired for the statistical modeling, relying on a total of 590 paired-values for density and 497 for SSA."

P9 L15 – Please provide the number of compared measurements to support of the significance test.
→ The number of compared measurements was 590 for density and 497 for SSA. We included that in the manuscript (see comment above).

P9 L16 – This differs substantially from Proksch et al (2015) where coefficients for SSA were not provided. This new equation requires no estimate of density from the SMP, which arguably is better if SSA is the target (minimizes bias from density coefficients and conversion from d0?). No action to take unless the authors wish to highlight the benefit of avoiding the conversion of L_ex to SSA.
→ We would agree with the reviewer that directly estimating SSA and not correlation length via the density as in Proksch et al. 2015, should lead to a better estimates (less errors). In the paper we simply pointed out this difference in the method by writing "Differing slightly from the one suggested by Proksch et al 2015, a regression of the from [Eq 2] was applied to estimate SSA …".

P9 L23 – An enhanced explanation of why the values in Figure 2 do not reflect the error/skill assessment in this section is needed. Related questions: Why does correlation improve when Eqn. (1) was trained on a different set of comparisons? Why was Eqn (1) was not just trained on this better alignment to begin with?

→ As written in an above comment on the same issue, we agree with the Reviewer and modified Section 5.1. In the revised version, values in Figure 2 (and the associated correlation analysis) are based on the same set of data and same re-alignment with the snow surface than the values taken for the statistical modelling Eq 1 and 2. Besides, our statement that using the MF-layer alignment leads to better correlation of values in Fig 2 was a wrong statement. Slightly better R2 coefficients are indeed found when using the snow surface than using the MF-layer for re-alignment (from layer alignment to snow surface alignment: R2 changes from 0.73 to 0.75 for density and from 0.81 to 0.82 for SSA, using Eq 1 and Eq 2 respectively). This makes sense as Eq 1 and 2 have been developed based on a snow surface re-alignment. This has been corrected in the revision and Section 5.1 is now consistent.

P8L29 – Remove one set of brackets around the Eqn.
→ done

P10 L2 – What was the statistical test that showed the boundary transition to be significant? If untested, consider removing the word 'significant'. See comments in the initiate statement about repeatability as well.

→ Boundaries were detected manually just from looking at the data, so there was no statistical test to identify them as well as to confirm that they are "significant". We deleted the work "significant" and it reads now, P10, L23: "The first step is to define which are the layers of interest, knowing that this method is only possible with layers that contrast well enough with their surrounding, so their boundaries can be easily identified by a rather sharp transition in the vertical profile of snow properties." We modified substantially Section 5.2 "Layer tracking", as described in a related comment above, so the method is better described now and can be repeated.

P10 L3 – Given that the boundaries were identified subjectively, will their heights be provided in the published dataset?

→ Heights of the tracked layers will be provided in the database of this study.

P15 L3 – I agree that the information is really useful to show the formation and evolution of these fine features. However, given that Figure 6b has no minor or major ticks for the initial date (Feb 22) it's fairly difficult to identity the feature. Could a label be provided for easy reference?

→ We prefer to leave the figures as is to avoid an emphasis on a single, annotated feature. Since the location is given exactly in the text and the x-axes of the subfigures are exactly the same, the birth of this layer could be easily taken from the SMP image above.

P23 L11 – I can confirm that the recalibrated density coefficients don't produce a best-possible estimates of snow density with our SMP for Arctic snow. Would be very interesting to combine datasets from multiple units to evaluate this uncertainty.

→ We agree that it would be very interesting to compare different sites to test the re-calibrations presented here.

P25L20 –Citation style should be a paraphrase.

Table 1 - List the number of measurements as a separate column. The large number of measurements is really smoothing to highlight! This will also be helpful in the future to frame comparison.
→ The number of measurements has been included in Table 1 (SMP: 100, Cutter: 15 profiles, IceCube: 13 profiles, Traditional: 11 profiles, Stability tests: 8 tests).

Figure 2 - Add N, R^2 and RMSE be added to these diagrams. Having a quantitative evaluation in the diagram provides a quick reference for the reader.
→ Done

Figure 4 – Please provide a colour legend for the grain type classifications even though they are standardized. Additionally, is it possible to provide sub-hatching for the hand hardness levels? It's challenging to determine the level past the first data.
→ Figure 4 has been modified accordingly.

Figure 6/9 – Has the SMP data been smoothed or aggregated? This does not appear to be mentioned in text but Figure 10 shows variability in SSA absent in Figure 9 at the 1 mm scale.
→ We used the same data with a resolution of 0.5 mm for the seasonal evolution plots as well as for the vertical profile plots (7 and 10).

---

## Author Comment (AC2) · 10 Apr 2020

**Response to Review 2**

We thank very much Reviewer 2 for his/her comments that help improving the manuscript. Please find below our point-by-point replies in blue color.

The paper presents the RHOSSA campaign focusing on snow density, SSA and sta- bility measurements over one winter in Weissfluhjoch, Switzerland. Modern methods such as SMP and IceCube are compared with traditional snow pit measurements and SNOWPACK modeling. Measurement results demonstrate how modern methods can increase temporal and vertical resolution in snow profiling compared with traditional measurements. This kind of data sets allow proper evaluation of modeling results, which is not possible using traditional measurements due to their poor temporal and vertical resolution. The main result is the recalibration of Proksch et al. 2015 model for deriving SSA and snow density from SMP data.

The snow stability part is a bit disconnected from the main text, which focuses on SSA and density. The authors could consider dropping the stability measurements.
→ We understand the concern raised here. However, although the mechanical properties are not analysed in as many details as for the structural properties, we think providing the complete dataset, including mechanical properties, can be very useful for other studies, as studies related to avalanches for example. It is important to keep in mind that traditional snow observations have a long tradition in avalanche research, which supports daily snow observations in alpine regions (such as Switzerland or France). And since nowadays stability predictions become feasible from high-resolution density profiles we definitely want to keep it. The full dataset is made available through a doi given in the paper.

p4r4 Section 6-> Section 5
→ Corrected

p4r12 Degrees missing from coordinates.
→ Corrected

p6r15 The snowpack was sampled with 3 cm resolution. What did you do with layers thinner than 3 cm? This explains why the 22 Feb layer is "only diffusely reported in the IceCube data" (p16 r17), if it is mixed with grains from other layers. How did you sample the MF layers? They are very difficult to get into sample holder without breaking them. Were the low density layers compacted to avoid measuring the sample holder?
→ Density and SSA profiles were recorded at regular height intervals of 3 cm, without considering the layering (we did the same for the SMP data with a vertical resolution of 1 mm and with the tomography data with a resolution of 18 μm). Using regular vertical grids and not following defined layers allows comparing data from different measurements and simulations, solely based on height (objective), without the need to identify layers (which can be subjective). We agree that a vertical resolution of 3 cm can lead to sampling in layer transitions leading to a more diffuse picture of the density or SSA profile. The MF layers were not too difficult to sample in our case (not overly dense) and procedure was the same as for other layers. Unfortunately, we are not aware of the method of compacting low-density layers to avoid measuring sample holder.

p8r8 Why exactly 1.2 ∘C?

→ The question of the impact on simulations from not considering the phase of precipitations cannot be answered straight away as we currently do not have observations permitting a proper attribution of precipitation phase at Weissfluhjoch. However, in preparation of the first SnowMIP around 2000, a dataset including the phase (liquid/solid, no mixed precipitations) and based on visual observations of the current weather could be constructed. The observations led us then to use a threshold of 1 °C. The threshold of 1.2 °C for Automatic Weather Station located above ~1000 m a.s.l. was introduced for operational use and proved to be well suited for Switzerland and Weissfluhjoch in particular (see  Schmucki et al., 2014)

Along the period considered in this paper, there were no major precipitations associated with air temperatures above 0 °C though.

In summary, this threshold plays no role in the context of this study and it would be out of scope to discuss it further in the text. Nevertheless, we reformulated slightly that sentence in Section 4 of the paper.

p9r10 What is the justification for selecting different method for matching the profiles here than later in the paper (p9r24)? If re-aligning profiles using the MF layer resulted in "better correlation between estimates from SMP and snow pit measurements", why didn't you use the same method here to derive the parameters?

→ We thank the reviewer for pointing out this issue in the paper. We agree on the confusion about the alignment methods used in Section 5.1. The revised version of the paper was modified so that alignment done for the statistical modelling (Eq 1 and 2) and, later, to compare cutter/IceCube data and SMP data (Fig 2) is *the same and based on the snow surface*. Using the snow surface to re-align co-located and co-temporal profiles is the more convenient and systematically applicable method that can be done by others in the same way (unlike using a specific layer of the snowpack).

Besides, our statement that using the MF-layer alignment leads to better correlation of values in Fig 2 was erroneous. Slightly better R2 coefficients are found when using the snow surface than using the MF-layer for re-alignment (from layer alignment to snow surface alignment: R2 changes from 0.73 to 0.75 for density and from 0.81 to 0.82 for SSA, using Eq 1 and Eq 2 respectively). This makes sense as Eq 1 and 2 have been developed based on a snow surface re-alignment.

Finally, the re-alignment based on the MF-layer is now only used in the Result part, for time-series plotting purposes for which snow surface alignment is not relevant as profiles are not co-temporal anymore (evolution of snowpack height over the season). Modifications concerning the alignment method were done throughout the paper, especially:

- Figure 2 has been redone, based on data re-aligned using the snow surface
- R2 coefficients associated to Figure 2 have been modified. They are slightly better than the previous version (from layer alignment to snow surface alignment: R2 changes from 0.73 to 0.75 for density and from 0.81 to 0.82 for SSA, using Eq 1 and Eq 2 respectively). This actually makes sense as Eq. 1 and 2 have been developed from data aligned with the snow surface.
- Section 5.1 reads now, p10, L6: The performance of the new parametrizations compared to the original parametrizations of Proksch2015 is presented in Figure 2. This plot shows the observed density from cutter measurements against the SMP-derived density obtained from Eq. (1) and from Proksch2015 for the 15 days for which both data are available (same data as used for the statistical

modelling). Similarly, the observed SSA from IceCube measurements are presented against the SMP-derived SSA from Eq. (2) and from Proksch2015 for the 13 days for which both data were available (again, same data as used for the statistical modelling). To do so, and as done for the statistical modeling, SMP-derived properties were averaged over 3 cm resolution and SMP and snow pit profiles of the same day were re-aligned with the snow surface and cropped to the length of the shortest profile. "

- In the introduction to the Result part, p12, L3, we explained further the choice of the MF-layer alignment to do temporal plots: "To present the evolution of profile properties with time, vertical profiles presented in the following were re-aligned such as z = 0 cm corresponds to the height of the upper boundary of the MF-layer (i.e. the 20151202-boundary). Choosing this layer as a height reference leads to a qualitatively better match than by simply taking the ground as reference (the field site ground at WFJ is uneven)."

p9 The model parameters are derived from IceCube measurements. Later (e.g. Fig 11) you show that there are big differences between IceCube and tomography measurements. Please comment on the accuracy of SMP-derived SSA values.
→ Comparisons between SSA measurements are described in Section 7.3. The SMP-derived SSA values inherits from 1/ the accuracy of the IceCube measurements (since the SMP-derived SSA values come from a fit (Eq 2) of the IceCube data) and 2/ the quality of the statistical model (how good is the fit).
Concerning 2/, the quality of the model is described in Section 5.1 and in Figure 2 (scatter plot). To describe further correlation of values in Fig 2, we included the RMSD values. P10, l14 now reads: "Applying a simple linear correlation between rho_cutter and rho_smp, a R2 coefficient of 0.87 and a root-mean square deviation (RMSD) of 34 kg m$^{-3}$ are found when using Eq. (1) against a R2 of 0.75 and a RMSD of 69 kg m$^{-3}$ when using the parametrization of Proksch et al. (2015). Between SSA_ic and SSA_smp, a R2 coefficient of 0.82 and a RMSD of 7 m$^2$ kg$^{-1}$ are found when using Eq. (2) against a R2 of 0.65 and a RMSD of 14 m$^2$ kg$^{-1}$ when using the parametrization of Proksch et al. (2015)."
Also, in Section 7.3 (line 14 page 24), we raise the point that the present statistical model used to derive SSA from SMP measurements fails to reproduce the high SSA values of newly-deposited snow, and that this could be because of their under-representations (only one day) in the IceCube dataset used for calibration.
Point 1/ is mentioned in Section 7.3 such as "First, we recall that density and SSA derived from SMP data were obtained to best match results from the cutter and IceCube measurements, so they necessarily inherit their performances" (p 23, L. 27). This implies that any discrepancies between IceCube and tomography data will necessary be also found between SMP-derived data and tomography data.

p11r2 choose->chose
→ modified accordingly

p11r20 caption->panel
→ modified accordingly

p22 Fig 11. The difference between SSA derived from SMP and tomography varies between different layers. Do you think the snow structure (grain type) has something to do with that? Should the SSA model be calibrated separately for different grain types?

And why are there big differences between IceCube measurements and SMP, if IceCube data was used in the fitting, shouldn't they agree better?

→ From Figure 11, we think that the variations in the differences between SSA derived from SMP and tomography depend more on the range of SSA values considered, rather than on the layers considered and so on the snow structure. Indeed, the quality of Eq 2 is better for some SSA ranges than other. In particular, looking at Figure 2b at SSA values below 20 $m^2$ $kg^{-1}$, we see that most of SMP values are slightly overestimated compared to IceCube values (cloud of values slightly below the 1:1 curve). Back to Figure 11, this bias clearly appears for most layers, for which SSA values are all mostly below 20 $m^2$ $kg^{-1}$. We add a sentence in the paper about this comment, which reads, P24, L7: "Note that one major discrepancy between IceCube and SMP-derived SSA comes from that the calibration used (parameterization) leads largely to an overestimation of the SSA values below about 20 $m^2$ $kg^{-1}$ by the SMP compared to IceCube (see Figure 2b, data cloud is mostly located below the 1:1 curve). This can be clearly seen in our results (Figure 9, 10, and 11) since a large part of the snowpack shows SSA values below 20 $m^2$ $kg^{-1}$."

→ Regarding the differences observed between SMP estimates of SSA and IceCube, they are directly link to the quality of the prediction Eq 2. To explain why a better regression could not be obtained, we think one point is that some snow type might not be well captured because of the under-representation in the IceCube measurements for some snow types, such as fresh snow in our case (this was the case of only 1 day on measurement for which fresh snow was measured in the first cm of the snowpack). To improve that, the calibration dataset should be extended so that all snow type is rigorously covered.

→ Regarding the grain type: a large part of the motivation of this work is making a step away from (subjective) indices. Thus re-introducing grain-type dependent calibration coefficients is, from our perspective, the wrong way to go. But it is true that the microstructure has an impact on the performance of the calibration model. This is the reason that only by introducing the SMP parameter L into the model, a significant improvement of the calibration (in particular in depth hoar) could be made over the old approaches of just using the median of the SMP force. Similar things are expected to happen for other snow types. The fact that the SSA point cloud in Fig 2 is not straight but slightly curved further supports that the present calibration model is still missing essential physics.

p23 Fig 12. Please add SNOWPACK profiles as well.
→ SNOWPACK simulations were added in Fig 12.

---

## Author Comment (AC3) · 10 Apr 2020

**Response to Review 3**

We thank very much Reviewer 3 for his comments that help improving the manuscript. Please find below our point-by-point replies in blue color.

General comments: The paper highlights results from a winter field campaign based out of the well-known WFJ site in Davos. The authors present a temporal analysis of snow microstructure and mechanical properties using state-of-the-art instruments that all have their advantages and limitations. Of particular relevance, repeated SSA and resistance measurements using an IceCube and the SMP are presented and com- pared against SNOWPACK simulations. The originality of the paper reside in a new calibration for the V4 of the SMP that will be indeed useful for international users such as my own group.

Overall, the paper is clearly written, with a very thorough analysis that certainly is worthy of publications. The expertise and reputation of the author's list is obviously excellent. I however, have several comments and questions that I would like to see addressed from my own perspective of being a SNOWPACK user in the Arctic with our own SMP and IRIS instrument since I think some elements need stronger analysis or at least physical explanations from the results presented in the paper given that very important science questions remain open.

Specific comments: In general terms, using SNOWPACK is not trivial. Yes the model can run virtually anywhere, especially in Switzerland where it was developed but cer- tainly harder elsewhere. A realization we came with as being users since 2002 is that the model remains very sensitive to 1) forcing dataset, 2) soil configuration and 3) ob- viously the internal physic calculations of microstructural elements that have changed from version to version over the years. For instance, a bias is observed in Canada on snow depth as a function of precipitation rate; or again bias in microstructure are not the same given the metamorphic process in place (kinetic vs equilibrium). Section 4 of the paper present the model in very general terms, I would suggest modifying this section to: SNOWPACK configuration where the authors would list: better description of the meteorological forcing dataset; soil configuration (type, roughness, how many soil layers?). There is also no mention of the spin up? Was the simulation initiated with a snow profile? It is obvious form the author list that the simulation is more than likely to be well parameterized, simply that I think there are more and more SNOWPACK users aware of potential problems, so more details on the simulation configuration I think would be very beneficial.

→We agree with the reviewer that the configuration of SNOWPACK needs a few more additions regarding initialisation, soil, etc. We adapted Section 4 accordingly. However, it is out of scope to present a detailed description of either the model or the data set in this study that has a quite different focus. Instead we will refer to Wever et al. (2015) that contains all information needed, except for the new settlement scheme. Indeed, and unfortunately, that part has only been presented in the frame of EGU 2011 but was not published yet. We are planning to do so soon. In addition, the dataset will be made available on Envidat upon acceptance.

Page 2, Line 8: 'spatially consecutive'...what is meant exactly? A clarification be appreciated. I assume the snowpit in such a confined space is useful for time-series, to avoid any variability due to spatial variability processes.

→ By "spatially consecutive" we meant those snow pits are dug consecutively during the season. We modified the sentence, P2, L6: "Regular snowpack monitoring programs rely on weekly to bi-weekly manual observations and measurements, by digging snow pits along a profile line in the (nearly) homogeneous observation area."

Page 2, Line 18: I would argue to add as a more general term the importance in surface energy balance, which in turn plays a critical role in freeze-thaw cycles for example. So the importance for large scale processes.

→ We agree and modified the sentence as follows: "It is defined by the ice/air interface surface area divided by the snow mass, which is inversely proportional to the optical grain size. SSA drives many snow processes as metamorphism, radiation interaction, air flow, chemical reactions and thus plays an important role in many large scale processes such as surface energy balance (e.g. Domine et al. 2007).

Page 3, Line 5: . . .change to gap in temporal resolution
→ Here we meant the gap in temporal and spatial resolution. We modified accordingly.

Page 3, Line 17-18: This was the whole idea behind the Snow Grain workshop held several years ago. Would the authors consider revisit some of the data?
→ We agree with the Reviewer that it will be a good idea to work with the data of the Snow Grain workshop. As far as we know, there are no plans in this direction for now.

Page 3, Line 29: how were selected the sites? It is mentioned that site were chosen on 'selected locations' but we all know site selection is critical. Some details on how the sites/samples were chosen be appreciated.

→ We agree and added more details on the selected locations, which were chosen to monitor the bottom part of the snowpack during the winter, i.e. the snow located around the persistent crust (MF-layer) and the weak layers (DH-layer and FC-layer). Once during the season we extend our sampling up to the slab on top of the FC-layer (including the RG-layer). These details are now provided in the paper so it reads now P3,L31: "occasional profiles of the 3D microstructure at 18µm vertical resolution from X-ray tomography (not full-depth, only on selected heights in the snowpack, mostly focusing on defined layers of interest)". Besides, we also include P4, L24: "X-ray tomography measurements of extracted, decimeter-sized samples were occasionally performed six times during the season at selected locations to image some defined layers of interest and allow further comparisons."

Table 1: add units to the measured/derived properties.
→ done

Page 5, Line8: ECT are extended column test, not extended compression test.
→ corrected

Section 3.2.: What was used to weigh the density cutter?
→ A digital scale was used to weigh the density cutter.

Page 7, Line 8: The 10% ucertainty is for IceCube or DUFISSS? IceCube was used, but the reference provided is for DUFISSS. What is the published accuracy of IceCube?
→ We provide here the uncertainty from Gallet et al. 2009, i.e. DUFISSS, assuming that uncertainty for IceCube is likely to be the same as the latter has been developed directly based on DUFISSS. As far as we know, we are not aware of a study specifying the accuracy for IceCube specifically.

Page 8, Line 8: I know the 1.2C threshold is used, likely well parameterized for WFJ. However I assume mixed precipitations are possible, what uncertainty can arise from such cases? A study by Ding et al. (2014) suggest that precipitation type are not only a factor of Tair, but also altitude and relative humidity. So how precise, at WFJ is precipitation phase parameterized?
→ The question of the impact on simulations from not considering the phase of precipitations cannot be answered straight away as we currently do not have observations permitting a proper attribution of precipitation phase at Weissfluhjoch. However, in preparation of the first SnowMIP around 2000, a dataset including the phase (liquid/solid, no mixed precipitations) and based on visual observations of the current weather could be constructed. The observations led us then to use a threshold of 1 °C. The threshold of 1.2 °C for Automatic Weather Station located above ~1000 m a.s.l. was introduced for operational use and proved to be well suited for Switzerland and Weissfluhjoch in particular (see Schmucki et al., 2014).
Along the period considered in this paper, there were no major precipitations associated with air temperatures above 0 °C though.
In summary, this threshold plays no role in the context of this study and it would be out of scope to discuss it further in the text. Nevertheless, we reformulated slightly that sentence in Section 4 of the paper.

Section 5.1: Our group is also doing just that with our own SMP this winter. Our concern is, that we are working on deriving a SMP(lc) method based on vertical 'z- axis' measurements from the SMP, with IceCube and density cutter that have a strong 'y-axis' component. We are asking ourselves if the SMP 'F' and 'L' parameter would be the same if we were to conduct a SMP profile in the 'y-axis' (i.e. in H instead of V)... From an anisotropy point of view, I think we can expect them to be different. Also we have an IceCube that includes a very thin layer being samples, with a density cutter that include a lot more snow... We are dealing with different scale, yet trying to correlate them together, I am fully aware that for now, this is the way to do it. Simply that I'd be happy to hear the authors ideas on this offline.

Section 6.1.: the problems linked to the vapor flux parameterization behind the growth of depth hoar is well known (Domine et al, 2019; Gouttevin et al., 2018). I'm also aware of the current work done in author's lab to correct that problem. Given the temperature and snow depth stated, yes I'm not surprise to see presence of depth hoar. Although, I'm pleased to see that SNOWPACK seems to react quite well to this, especially when I'm looking at Figure-7 where the depth hoar layer is indeed corresponding to a reduction in density as can be expected. This was a problem, that now looks much better. So my question is: did the authors used a different metamorphism parameterization to reach this? Or the standard version online was used without further modification?
→ Thank you for mentioning this. We added a sentence in the text to draw the reader's attention to it. However, neither changes nor adaptations to the metamorphism scheme

of SNOWPACK were implemented to reach this in our study. Indeed, whenever a deep depth hoar layer develops at the bottom of the snowpack at Weissfluhjoch, the resulting lower density of that basal layer is reasonably captured by SNOWPACK. For example, see winters 2015, 2005, and 2002 in Wever et al, (2015). A close inspection of the newly added Figure 7 however reveals that SNOWPACK still systematically underestimates the density of the slab while the density of the base is still overestimated the DH base. This effect is minor for this alpine snowpack but may still be emphasized for more extreme DH formation. We added some discussion on the performance of SNOWAPCK to simulate depth hoar layer point in Section 7.2

Figure-4 would be much easier to read with a legend.
→ A legend has been added to Figure 4.

Page 16, Line 2: Why does SNOWPACK overestimate the density of the DH layer? Is it because of the absence of vapor flux from the ground leading to the underestimation of the SSA?
→ The overestimation of density of the DH-layer from SNOWPACK is probably, at least partly, linked to the absence of the MF-layer in the simulations (not formed), Without this dense, stiff layer, we think that more load might have been transferred to the DH-layer leading to more densification. More work would be needed (we could force the simulation of a crust and compare simulations with and without it for example) to investigate the origin of this overestimation. Concerning SSA, SNOWPACK underestimates values in overall, so for all snow types and not more particularly for the DH-layer. The main cause is thus likely not only linked to an effect close to the ground, which would affect mostly basal layers, but maybe to the SSA parameterisation scheme implemented in SNOWPACK. However, as pointed out in Section 7.3, it is difficult to evaluate the SSA simulations in details because of the significant inter-measurement deviations observed. Dedicated works on that topic would be necessary.

Section 6.3: When using IceCube, it is very hard to sample properly depth hoar by the simple nature of the thickness of the hoar layer vs the sampler size. Any sampling difficulties were encountered using IceCube in these conditions?
→ We did not encounter major difficulties to sample the depth hoar layers that were indeed made of rather large crystals but also rather dense (around 300 kg/m3). The latter might have contributed to facilitate the sampling. For the layers that are difficult to sample, including depth hoar but also in our case fresh snow or crust, we might had to repeat the sampling and measurement so that consistent, reliable values were obtained.

Section 7.1, Lines 9-10: I would argue that yes there is a range, but it remains alpine where the processes governing stratigraphy, energy transfer is a different world from what we find in the Arctic, or even in other alpine regions of the world. I would argue to state that the snowpack offered a wide range of alpine snow conditions.
→ We agree and modified the sentence accordingly such as P21, L3: "Yet, the snowpack monitored over winter 2015-2016 offered a wide range of alpine snow type and property variations throughout the season."

Section 7.2.: With a snowpack having a temperature gradient important enough to lead to the formation of a depth hoar layer, can expect to have a decent variability in temperature vertically obviously. But, the effect of changing temperature as the SMP

travels through snow is not discussed. I know the authors are aware of this problem, can they confirm this was not an issue in this environment?

→ We are indeed aware of this problem of the influence of temperature on the penetration resistance signal. We did not observed any anomaly or drift in values that could be related to this effect on the technical side. On the scientific side, for the present calibration we did not take into account any temperature dependence of the calibration parameters. In principle, such an extended analysis seems readily feasible from the present dataset by re-evaluating the structural data together with the (relatively robust predictions) of temperatures from SNOWPACK. It remains unclear though if temperature trends could be statistically discerned from microstructural (snow type) effects. But the present calibration must be considered as an average over all naturally occurring temperatures in the snow profile.

Page 24, Line 12: I think it is more a problem of the laser hitting the side of the sampler rather than the bottom, but this is a small detail.

→ We thank the Reviewer for the comment and would be happy to exchange more on this issue.

Again, this is a very nice contribution made by a very solid team at a site internationally known. I would suggest my comments to be minor, and would be happy to see this work published after the comments above are addressed.